# UEx-L-Eddies: Decadal and global long-lived mesoscale eddy trajectories with coincident air-sea CO₂ fluxes and environmental conditions

Daniel J. Ford [1], Jamie D. Shutler[1], Katy L. Sheen[1], Gavin H. Tilstone[2], and Vassilis Kitidis[2]

[1] Centre for Geography and Environmental Sciences (CGES), University of Exeter, Penryn, UK.
[2] Plymouth Marine Laboratory, Plymouth, UK

*Correspondence to*: Daniel J. Ford (d.ford@exeter.ac.uk)

**Abstract.** Mesoscale eddies are prevalent features within the global ocean that modify the physical, chemical and biological properties as they move and evolve. These modifications can alter the air-sea exchange of $CO_2$, and therefore these features may be hotspots for enhanced or reduced $CO_2$ uptake compared to the surrounding environment. The understanding of the global and regional effect of mesoscale eddies on ocean $CO_2$ uptake is however limited and largely based on single eddies or small regional subsets. Here, we provide a global dataset of 5996 long lived eddies trajectories (lifetimes greater than a year) with corresponding air-sea $CO_2$ fluxes all tracked using a Lagrangian approach between 1993 to 2022. The trajectories comprise 3244 anticyclonic ('warm core') and 2752 cyclonic ('cold core') eddies and the dataset provides the environmental conditions, including the $CO_2$ fluxes, within and outside each eddy. The dataset refines a previous regional methodology with a focus on climate quality environmental parameters and uses a global neural network for estimating the fugacity of $CO_2$ in seawater ($fCO_{2\ (sw)}$) along with a comprehensive air-sea $CO_2$ flux uncertainty budget. These refinements provide a robust foundation for studying the modulation of air-sea $CO_2$ fluxes by mesoscale eddies. As an example use of the dataset, we investigate the role of mesoscale eddies in modifying the global and regional air-sea $CO_2$ fluxes, by comparing the eddy driven air-sea $CO_2$ flux to that of the surrounding environment. We find that globally, long-lived anticyclonic eddies enhanced the $CO_2$ sink by $4.5 \pm 2.8$ % (95 % confidence), while long-lived cyclonic eddies reduce the $CO_2$ sink by $0.7 \pm 2.6$ %. Collectively, the long-lived eddies indicate an enhancement of the ocean $CO_2$ sink by $2.7 \pm 1.1$ Tg C yr⁻¹. Propagating the air-sea $CO_2$ flux uncertainties was found to be a key component needed to fully understand apparent differences between previous regional and global studies. The long lived eddies (UEx-L-Eddies) dataset is available on Zenodo at https://doi.org/10.5281/ZENODO.16355763 (Ford et al., 2025).

## 1. Introduction

Mesoscale eddies are known to affect the physical, chemical and biological properties of the oceans (Dufois et al., 2016; Frenger et al., 2013; Laxenaire et al., 2019; Li et al., 2025; Nencioli et al., 2018; Orselli et al., 2019b, a; Pezzi et al., 2021). These rotating bodies of water have radii on the order 100 km, lifetimes from a few days to multiple years, and can transit ocean basins transporting distinct water masses within them (Chelton et al., 2011; Pegliasco et al., 2022b). Eddies generally fall into two categories; (1) anticyclonic and (2) cyclonic. Anticyclonic eddies are associated with high pressure centres, clockwise rotation in the Northern Hemisphere (or anticlockwise in the Southern Hemisphere), warmer sea surface temperatures (SST), and a depression of isopycnals (and downwelling of water within the eddy core). Whereas cyclonic eddies are generally the opposite; low pressure centres, anticlockwise rotation in the Northern Hemisphere (or clockwise in the Southern Hemisphere), cooler SSTs, and an elevation of isopycnals (and upwelling in the eddy core). During their lifetimes, these eddies can alter the air-sea $CO_2$ exchange through their modification of the ocean and atmospheric properties. As the $CO_2$ solubility in seawater is highly temperature sensitive, the $fCO_{2\,(sw)}$ in anticyclonic eddies could theoretically be elevated and therefore the features may act as a weaker $CO_2$ sink or stronger $CO_2$ source compared to the surrounding environment. Conversely the opposite may be true for cyclonic eddies, with reduced $fCO_{2\,(sw)}$, and increased capacity to act as a stronger ocean $CO_2$ sink. But mesoscale eddies are complex dynamic features, and these generalisations may not always apply as their response will always be dependent upon the ocean basin conditions where the eddy formed and through which the eddy moves, along with how they evolve and interact with that ocean water and the atmosphere. For example, Chen et al. (2007) identified a cyclonic eddy acting as a weaker $CO_2$ sink compared to the surrounding environment due to upwelling of $CO_2$ and nutrients within the eddy core. Orselli et al. (2019b) showed six anticyclonic Agulhas eddies that were acting as a stronger $CO_2$ sink (than the surrounding water) during Austral winter. Pezzi et al. (2021) identified an anticyclonic eddy acting as a strong $CO_2$ source in the Southwestern Atlantic. Whereas, through using a biogeochemical model, Song et al. (2016) suggested that these eddy modifications may have seasonal differences, whereby anticyclonic (cyclonic) eddies acted as stronger (weaker) $CO_2$ sinks in summer, but stronger (weaker) sources in winter.

Despite the abundance of mesoscale eddies, previous studies generally investigate singular eddies (Chen et al., 2007; Jones et al., 2017; Pezzi et al., 2021) or a regional subset of eddies (Ford et al., 2023; Orselli et al., 2019b; Song et al., 2016) and their effect on the air-sea $CO_2$ flux. Thus, the global cumulative effect of all types of eddies on the air-sea $CO_2$ flux is still under investigation. Ford et al. (2023), used a Lagrangian tracking approach and suggested that long-lived (lifetimes greater than one year) mesoscale eddies enhanced the air-sea $CO_2$ flux in the South Atlantic Ocean by ~0.05 Tg C yr$^{-1}$ (~0.08%). Guo and Timmermans (2024) used a spatial and timeseries decomposition to extract the mesoscale flow impact on the air-sea $CO_2$ fluxes globally, and estimate a small integrated effect of 0.72 Tg C yr$^{-1}$ (compared to global ocean uptake of ~2.9 Pg C yr$^{-1}$). However, this result may include mesoscale signals not related to mesoscale eddies (Guo and Timmermans, 2024).

Li et al. (2025), using a method that tracked individual eddies similar to Ford et al. (2023), showed that mesoscale eddies within the Kuroshio and Gulf Stream western boundary currents could enhance the $CO_2$ sink by $28.34 \pm 9.41$ Tg C yr$^{-1}$.

60

In this paper we produce a global dataset of long lived (defined as lifetimes greater than one year) mesoscale eddies (N = 5996; radii > 30 km) and their associated air-sea $CO_2$ fluxes tracked in a Lagrangian mode between 1993 and 2022. The methodology refines the approach described in Ford et al. (2023), using a global neural network approach and published tools which are also used to generate one ocean carbon sink dataset submission to the annual Global Carbon Budget assessments (Friedlingstein et al., 2025). Following recommendations for global ocean carbon assessments (Shutler et al., 2024) we prioritise the use of climate quality satellite data records (Embury et al., 2024; Sathyendranath et al., 2019) within the analysis. The uncertainties on the air-sea $CO_2$ fluxes are systematically assessed following the work of Ford et al. (2024a). These refinements provide a robust foundation to studying the modulation of air-sea $CO_2$ flux by mesoscale eddies, with an uncertainty budget. We demonstrate the use of the global dataset to assess regional and global air-sea $CO_2$ fluxes of long-lived eddies and to estimate their net impact on $CO_2$ uptake of the ocean.

## 2. Methods

Figure 1 shows a schematic of the implementation of the methodology within this study to estimate the air-sea $CO_2$ flux within mesoscale eddies.

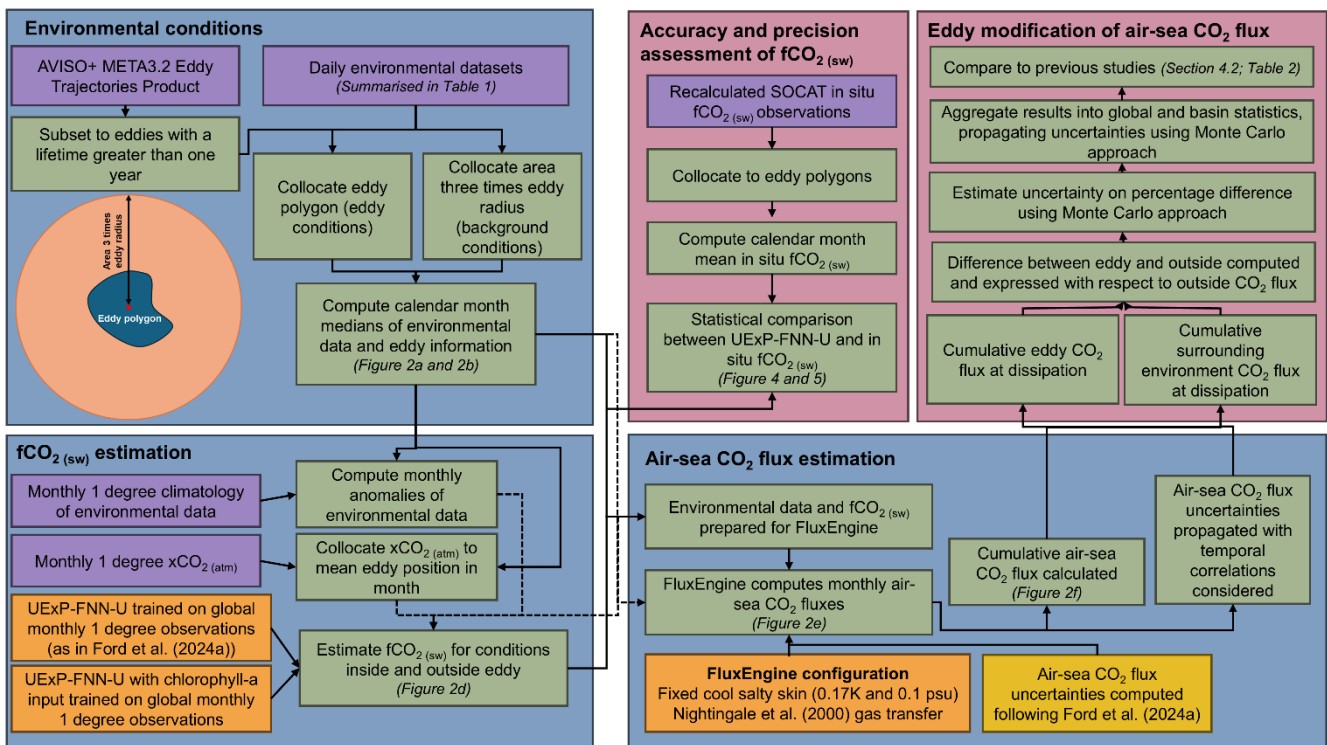

**Figure 1: Schematic showing the processing steps to estimate the air-sea CO₂ flux within long lived eddies (Blue box background). The pink background boxes indicate the analysis completed to evaluate the accuracy and precision of the dataset. In figure acronyms are: fugacity of CO₂ in seawater (fCO₂ (sw)), atmospheric dry mixing ratio of CO₂ (xCO₂ (atm)) and University of Exeter feed forward neural network with uncertainties (UExP-FNN-U).**

## 2.1 Satellite and reanalysis data

The importance of prioritising the use of climate data records to study long time series and the ocean carbon sink was highlighted in Shutler et al. (2024). We used the European Space Agency's climate change initiative (CCI) climate data records SST-CCI (v3; ~4 km; 1993 to 2022) for SST (Embury et al., 2024; Good and Embury, 2024) and the Ocean Colour CCI (OC-CCI) for the chlorophyll-a (chl-a) concentrations (v6; ~4 km; 1997 to 2022; Sathyendranath et al., 2019, 2023), with their respective per observation uncertainties (Table 1). The CCI-SST was bias corrected for a cool bias with respect to global SST drifters, representative of SST at 20 cm (~0.05K; Embury, 2023; Embury et al., 2024), which is used to provide an accurate estimation of fCO₂ (sw) (in section 2.3), and for the air-sea CO₂ flux calculation (in section 2.4).

We were unable to use the sea surface salinity (SSS) CCI climate data record for our application due to the 8 day temporal resolution of these data. We therefore used the Copernicus Marine Service GLORYS12V1 ocean reanalysis product for SSS (~9 km; 1993 to 2022; CMEMS, 2021; Jean-Michel et al., 2021), and the ocean mixed layer depth (MLD) as no climate data record is available for MLD. No climate data record is available for wind speed, therefore the Cross-Calibrated Multi-

Platform (CCMP) wind speed dataset (6-hourly; ~25 km; 1993 to 2022) was chosen (Mears et al., 2022; Remote Sensing Systems et al., 2022) which is often used for ocean carbon assessments (Ford et al., 2024a).

95    **Table 1: Summary of the environmental datasets and in situ observations collocated with the long lived mesoscale eddies.**

| Parameter | Units | Dataset | Temporal Resolution | Spatial Resolution | Reference |
|---|---|---|---|---|---|
| Sea surface temperature | Kelvin | ESA CCI-SST v3.0 | Daily | ~5km (0.05 degree) | (Embury et al., 2024; Good and Embury, 2024) |
| Sea surface salinity | Psu | CMEMS GLORYS12V1 | Daily | ~9km (0.08 degree) | (CMEMS, 2021; Jean-Michel et al., 2021) |
| Mixed layer depth | m | CMEMS GLORYS12V1 | Daily | ~9km (0.08 degree) | (CMEMS, 2021; Jean-Michel et al., 2021) |
| Chlorophyll-a | mg m$^{-3}$ | OC-CCI v6 | Daily | 4km | (Sathyendranath et al., 2019, 2023) |
| Wind speed | m s$^{-1}$ | CCMP v3.1 | 6 hourly | ~25km (0.25 degree) | (Mears et al., 2022; Remote Sensing Systems et al., 2022) |
| Sea level pressure | hPa | ERA5 | Monthly | ~25km (0.25 degree) | (Hersbach et al., 2019, 2020) |
| xCO$_{2\ (atm)}$ | ppm | NOAA-GML | Monthly | ~100km (1 degree) | (Lan et al., 2023) |
| fCO$_{2\ (sw)}$ | µatm | Recalculated SOCAT | Individual cruise observations | N/A | (Bakker et al., 2016; Ford et al., 2024d) |

## 2.2 Eddy Trajectories Atlas

The satellite altimetry based Mesoscale Eddy Product (version META3.2) as described in Pegliasco et al. (2022b, a), and distributed by the Archiving, Validation, and Interpretation of Oceanographic Satellite data (AVISO), was used to identify the trajectories of mesoscale eddies between 1993 and 2022. We extracted the eddy trajectories globally, that had a lifetime greater than one year, which gave 3244 anticyclonic eddies and 2752 cyclonic eddies for further analysis. The focus on these long-lived eddies was due to their presence likely exhibiting a larger influence on the air-sea $CO_2$ flux (e.g. Smith et al., 2023). Additionally, the selection was due to computational limitations in running the analysis for the extensive set of shorter lived eddies within the dataset. We are working to extend the analysis to shorter lived eddies but currently the focus remains on long lived eddies.

For each eddy trajectory, a daily position was provided along with a polygon shape that estimates the eddy shape and size from the altimetry-based data which can not overlap with land. These eddy polygons were used to extract a daily timeseries of the environmental data described in Section 2.1, where the daily conditions within the eddy were calculated (mean, median, standard deviation, interquartile range, maximum number of available data points, number of valid data points). This was repeated for the area surrounding the eddy, where we consider the 'area outside' to be a circle centred on the eddy but with three times the mean radius of the eddy and the area inside the eddy polygon itself removed. The chosen radii (of three times the mean radius) was used as Ford et al. (2023) showed that the results of their study were consistent when using a 'surrounding area criterion' between two and five radii.

Daily timeseries of conditions within and surrounding the eddy, were then converted to a monthly median timeseries using the daily median values. The daily median was chosen to reduce the impact of any potential outliers caused by any limited data coverage due to cloud cover in the chl-a record. The daily median and mean were generally consistent for the SST, SSS, MLD and wind speed fields as these are spatially complete fields.

## 2.3 $fCO_{2\,(sw)}$ neural network (UExP-FNN-U) and uncertainty

The monthly $fCO_{2\,(sw)}$ and air-sea gas fluxes were estimated using the methods and tools of the University of Exeter Physics Feed Forward neural network with uncertainties (UExP-FNN-U) which are routinely used to generate ocean sink data for the annual Global Carbon Budget assessments (Friedlingstein et al., 2025), and described in Ford et al. (2024a). The UExP-FNN-U approach estimates the $fCO_{2\,(sw)}$ based on in situ data that is considered representative of the subskin layer (~0.2 m water depth), which allows for an accurate air sea $CO_2$ flux calculation (Woolf et al., 2016; Section 2.4). The methods used are consistent with those in Ford et al (2024a), so only a summary of the method is provided here. The UExP-FNN-U is a two-step self-organising map (SOM) feed forward neural network (FNN) setup. The SOM splits the global ocean into 16 regions with a similar $fCO_{2\,(sw)}$, SST, SSS and MLD seasonal cycles. A FNN ensemble (10 FNNs for each region) was then trained with in situ monthly 1 degree $fCO_{2\,(sw)}$ observations from the Surface Ocean $CO_2$ Atlas (SOCAT; Bakker et al., 2016)

that have been recalculated to a consistent temperature and depth dataset (Ford et al., 2024d). The monthly 1 degree predictor variables of SST, SSS, MLD and the atmospheric dry mixing ratio of $CO_2$ ($xCO_{2\,(atm)}$), and anomalies of each with

respect to a long term monthly climatology were collocated to the in situ $fCO_{2\,(sw)}$. The FNNs consists of an input layer with nodes equal to the number of input predictors, a hidden layer with a varying number of nodes depending on a pretraining step and an output layer with a single node. The training data were split into a 95% training and validation dataset, and a 5% independent test randomly for each month ensuring the independent data were not clustered in one region. The UExP-FNN-U $fCO_{2\,(sw)}$ estimates are then typically used to estimate the global ocean $CO_2$ sink as described in Ford et al. (2024a).

To estimate the $fCO_{2\,(sw)}$ for each eddy the monthly median timeseries of the SST, SSS, MLD were provided to the UExP-FNN-U. The $xCO_{2\,(atm)}$ was calculated from the National Oceanic and Atmospheric Administration Global Monitoring Laboratory (NOAA-GML) monthly 1 degree fields (Lan et al., 2023) that were used within the neural network training. These $xCO_{2\,(atm)}$ fields were produced by calculating the monthly average of the $xCO_{2\,(atm)}$ for each latitude (~2.5 degree spacing), which were then interpolated to 1 degree and replicated for each 1 degree longitude. A distance weighted mean of

the nearest four pixels taken at the mean (centre) position of each eddy was used to estimate the monthly $xCO_{2\,(atm)}$. . Anomalies in SST, SSS, MLD and $xCO_{2\,(atm)}$ were calculated with respect to a 1 degree monthly climatology.

The uncertainties in the $fCO_{2\,(sw)}$ were calculated as described in Ford et al. (2024a). The $fCO_{2\,(sw)}$ uncertainty has three components: (1) the network uncertainty estimated as the two standard deviation of the 10 neural network ensemble, (2) the parameter uncertainty was the propagated input parameter uncertainties and was estimated using a lookup table and (3) the

evaluation uncertainty which was the evaluation with respect to the SOCAT observations (Bakker et al., 2016). All three components are combined in quadrature, assuming they are independent and uncorrelated (Taylor, 1997), to provide a total uncertainty (considered 95% confidence). The uncertainty components were calculated for each $fCO_{2\,(sw)}$ estimate.

Additionally, a second version of the neural network was run. This version included chl-a (and the chl-a anomaly) as a predictor and was used to produce a second estimate of $fCO_{2\,(sw)}$. Ford et al. (2022a) highlighted that the inclusion of more

representative biological parameters improved the regional estimation of $fCO_{2\,(sw)}$ in the South Atlantic Ocean. Therefore, this additional neural network output was generated using the same software used to create the UExP-FNN-U estimate of $fCO_{2\,(sw)}$ (Ford et al., 2024c) just with the added chl-a predictor. However, we note the limitation of this second $fCO_{2\,(sw)}$ estimate that uses chl-a. This dependency on optically derived remote sensing data (ie the chl-a data) means that it was limited to producing estimates after October 1997 (as routine ocean colour observations are not available before this date)

and it could not provide estimates during polar winter due to missing daily chl-a data (as the low light levels inhibit optical retrievals).

The neural network estimated $fCO_{2\,(sw)}$ were compared to recalculated SOCAT observations (Ford et al., 2024d; Goddijn-Murphy et al., 2015) within eddies to assess the accuracy and precision of the estimates. The individual cruise SOCAT observations are gridded (to monthly 1 degree) to provide the training and independent test data to the UExP-FNN-U, and

therefore these $fCO_{2\,(sw)}$ observations are not strictly independent. For each eddy trajectory, the ungridded SOCAT observations were collocated with the daily eddy polygon. The daily SOCAT observations that fell within the eddy were then aggregated into monthly mean $fCO_{2\,(sw)}$, which could be compared to the neural network monthly $fCO_{2\,(sw)}$. We calculated a series of statistics including the bias, root mean square difference (RMSD), slope and intercept of a Type II linear regression to characterise the differences between the neural network outputs and monthly mean SOCAT $fCO_{2\,(sw)}$. A

Type II linear regression was used as uncertainties are presented within both the in situ and neural network $fCO_{2\,(sw)}$ (Laws, 1997; York et al., 2004). As in Ford et al. (2021) weighted variants of these statistics were also calculated to capture the uncertainties in both sets of data (neural network output and the SOCAT in situ data), assuming a SOCAT $fCO_{2\,(sw)}$ uncertainty of 5 µatm (Bakker et al., 2016) and the calculated neural network total $fCO_{2\,(sw)}$ uncertainty.

## 2.4 Air-sea CO₂ flux calculations and uncertainties

The $CO_2$ flux calculations were performed using FluxEngine v4.0.9.1 (Holding et al., 2019; Shutler et al., 2016), using the "rapid" transport approximation (Woolf et al., 2016), at monthly time steps. The evidence continues to grow supporting the calculation of air-sea $CO_2$ fluxes with consideration of the vertical temperature gradients, which is supported by theoretical (Woolf et al., 2016), observation based (Dong et al., 2022b; Shutler et al., 2020; Watson et al., 2020), modelling (Bellenger

et al., 2023), and recently two in situ studies (Dong et al., 2024; Ford et al., 2024b). Therefore, the air-sea $CO_2$ fluxes were calculated using a bulk formulation that allows for the vertical temperature gradients to be captured. The calculations are consistent with the methods used to create the UExP-FNN-U dataset that is submitted to the annual Global Carbon Budget assessments (Friedlingstein et al., 2025), except here a simplified approach to determine the skin SST value is used.

The air sea $CO_2$ flux (F) was calculated as:

$$F = k_{600}\,(Sc/600)^{-0.5}\big(\alpha_{subskin}\,fCO_{2\,(sw,subskin)} - \alpha_{skin}\,fCO_{2\,(atm)}\big) \qquad (1)$$

Where k is the gas transfer velocity estimated from the monthly wind speeds and the Nightingale et al. (2000) gas transfer parameterisation. $\alpha_{subskin}$ and $\alpha_{skin}$ are the solubility of $CO_2$ at the base, and top of the mass boundary layer respectively, and were calculated as a function of SST and SSS (Weiss, 1974). $\alpha_{subskin}$ was calculated from the bias corrected CCI-SST SST and the CMEMS SSS. $\alpha_{skin}$ was calculated with the same datasets, but with a fixed cool (-0.17K) (Donlon et al., 1999) and

salty (+0.1 psu) skin effect. We used a fixed cool skin here, instead of the dynamic cool skin approach (that uses COARE 3.5; Fairall et al., 1996) as used within the UExP-FNN-U Global Carbon Budget submission due to the computation overhead needed to extract the additional environmental fields required for the calculations. This simplified approach has only a small effect on the global scale (Dong et al., 2022b), and therefore we do not see it as a limitation. $fCO_{2\,(atm)}$ was estimated for the NOAA-GML $xCO_{2\,(atm)}$, ERA5 sea level pressure (Hersbach et al., 2019) and the CCI-SST with a cool salty

skin following Dickson et al. (2007). $fCO_{2\,(sw,subskin)}$ was provided by the neural network $fCO_{2\,(sw)}$. The ERA5 sea level

pressure was retrieved from monthly 0.25 deg fields, using a distance weighted mean of the 4 closest observations to the mean monthly eddy position. None of the eddies considered were under sea ice (as the eddy detection data and algorithm cannot track in areas of ice), and therefore the term "1 – ice" which is generally included within Eq. 1 (to linearly scale the gas fluxes with sea ice concentration) has not been included.

The air-sea $CO_2$ flux uncertainties were calculated following the methods in Ford et al. (2024a), and consistent literature values for the uncertainties in the wind speed (1.9 ms$^{-1}$; 95% confidence; Mears et al., 2022a), salinity (0.2 psu; 95% confidence; Jean-Michel et al., 2021), $xCO_{2 (atm)}$ (0.4 µatm; 95% confidence; Lan et al., 2023) and gas transfer parameterisation (20%; 95% confidence; Woolf et al., 2019). The SST uncertainty was extracted from the daily CCI-SST dataset and were converted to monthly uncertainties assuming a five day temporal correlation (Ford et al., 2024a). The

uncertainties were calculated at the 95% confidence (or the 2 sigma).

The monthly mean daily flux of $CO_2$ (g C m$^{-2}$ d$^{-1}$) was multiplied by the number of days and the mean area of the eddy as provided by the eddy trajectories, in the respective month. The fluxes (Tg C mon$^{-1}$) were then added cumulatively to retrieve the net cumulative $CO_2$ flux for each eddy (Tg C). Collating the combined uncertainties requires careful consideration of their temporal correlations. Some uncertainties will be temporally decorrelated, and others have temporal correlations. We

used the assumptions made in Ford et al. (2024a), that the SST, SSS, wind speed, $xCO_{2 (atm)}$ and $fCO_{2 (sw)}$, and components dependent on these uncertainties, are temporally uncorrelated and are therefore propagated assuming they are independent (Taylor, 1997). Whereas, the remaining uncertainties that stem from the Schmidt number, solubilities and gas transfer parameterisation algorithm uncertainties are assumed temporally correlated and therefore are summed (Ford et al., 2024a). The air-sea $CO_2$ flux calculations and uncertainty estimates were computed for the two variants of $fCO_{2 (sw)}$. The

computations were also applied separately for the eddy and the area outside the eddy, assuming the same area coverage of the eddy for both calculations (i.e allowing the cumulative fluxes to be compared for the same area coverage).

### 2.5 Modification of air-sea $CO_2$ fluxes due to the existence of the eddy

As shown in Ford et al. (2023), the air-sea $CO_2$ flux into an eddy can be considered as two components: (1) the flux that would occur without the presence of the eddy and (2) the mesoscale modification of the flux through both oceanic and

atmospheric effects of the eddy presence. The flux that would occur without the eddy being present can be estimated using the conditions that are driving the air-sea $CO_2$ flux in the environment surrounding the eddy. This reference flux can be removed from the air-sea $CO_2$ flux calculated for within the eddy to indicate the mesoscale modification of the flux due to the existence of the eddy, which was converted to a percentage change with respect to the surrounding environment $CO_2$ flux, following Ford et al. (2023).

The eddy modification of the air-sea $CO_2$ flux was calculated for each individual eddy, and then the median percentage modification was estimated for global and regional subsets, due to the lower sensitivity to outliers. We repeat the percentage

change calculations in a Monte Carlo uncertainty propagation approach to evaluate the full extent of the uncertainties, whereby the eddy modification flux was perturbed within their uncertainties (95%) 1000 times independently (i.e., assuming the individual eddy flux modification uncertainties are uncorrelated). The two standard deviation value of the resulting ensemble was taken as the 95% confidence on the median percentage change for the global or regional subsets due to the uncertainties.

## 3. Results

### 3.1 Geographical distribution of mesoscale eddy cumulative air-sea CO$_2$ flux

In total 5996 eddies were tracked and their air-sea CO$_2$ flux estimated, which comprised 3244 anticyclonic and 2752 cyclonic eddies between 1993 and 2022 (Figure 2). The geographical distribution of the cumulative air-sea CO$_2$ flux into both eddy types generally followed the global distribution of air-sea CO$_2$ fluxes. The temperate regions showed eddies with strong CO$_2$ sink characteristics over their lifetimes, whereas eddies in the subtropical showed weaker CO$_2$ sinks, or even CO$_2$ sources. Regionally the Indian Ocean showed stronger CO$_2$ sinks associated with anticyclonic eddies when compared to the Atlantic and Pacific Oceans (Figure 2a). The South Pacific showed anticyclonic eddies acting as weaker CO$_2$ sinks compared to the North Pacific and had more eddies acting as CO$_2$ sources. Notable regions where cyclonic eddies were acting as strong CO$_2$ sinks are within the Indian Ocean, and Northwestern Atlantic Ocean (Figure 2b). Cyclonic eddies in the South Pacific tended to act more as CO$_2$ sources than sinks (Figure 2b). The Southern Ocean showed the anticyclonic and cyclonic eddies acting as either weak CO$_2$ sinks or weak CO$_2$ sources (Figure 2).

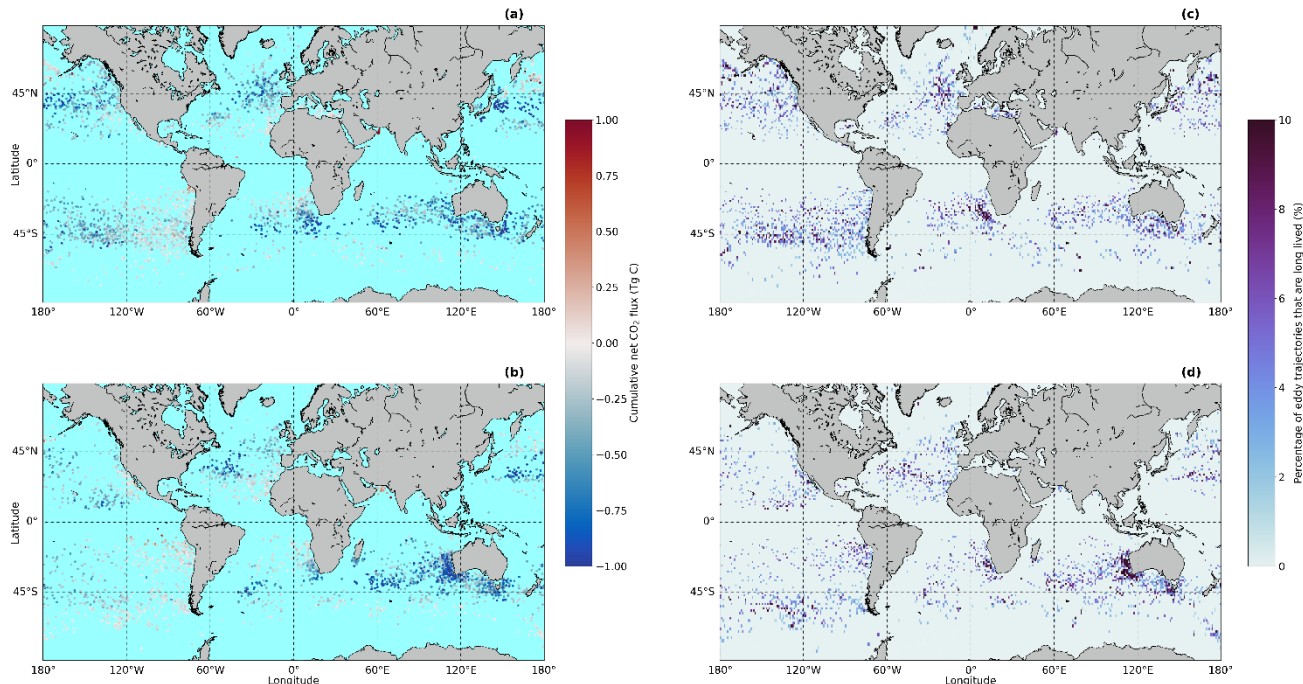

Figure 2: (a) The cumulative air-sea $CO_2$ flux into the anticyclonic eddies where the scatter points are plotted at the formation location of each eddy. (b) same as (a) but for cyclonic eddies. (c) The percentage of long lived anticyclonic eddy trajectories compared to all eddy trajectories that form in 1 degree by 1 degree regions. (d) same as (c) but for cyclonic eddies. Basemap from Natural Earth v4.0.0 (https://www.naturalearthdata.com/). Supplementary Figure S1 shows the equivalent of (a) and (b) in Tg C d⁻¹ to remove the differences in eddy lifetime.

## 3.2 Example eddy trajectory

Figure 3 shows an example of an eddy trajectory in the North Pacific Ocean that was selected due to the ~3 year lifetime, that highlights the seasonality and variability of the environmental data, the $fCO_{2\,(sw)}$ and associated air-sea $CO_2$ fluxes with the uncertainties shown. Over the three years the eddy moves around a relatively small region within the subpolar region (Figure 3c). Within the eddy, an expected SST seasonal cycle was present (Figure 3a), along with an interannual variability within the SSS timeseries (Figure 3b). The estimated $fCO_{2\,(sw)}$ also highlighted a clear seasonal cycle with higher $fCO_{2\,(sw)}$ in the winter months, and lower $fCO_{2\,(sw)}$ in the summer (Figure 3d). The eddy exhibited a period of strong $CO_2$ outgassing during winter, followed by a small $CO_2$ sink within the summer months (Figure 3e). When cumulatively summed, the air-sea $CO_2$ fluxes indicate that the eddy outgassed $CO_2$ over its lifetime, but clearly this outgassing was not year-round (Figure 3f). The example eddy illustrates the available data that could be used to evaluate the driving mechanism that are affecting the $fCO_{2\,(sw)}$ and air-sea $CO_2$ fluxes over the eddy's lifetime.

### 3.3 UExP-FNN-U fCO$_2$ (sw) compared to SOCAT observations within eddies

The UExP-FNN-U was trained on a global dataset of fCO$_2$ (sw) and so it is important to assess its performance within eddies providing some level of confidence that the eddy variability is being correctly captured. The within eddy accuracy and precision estimates between the SOCAT in situ observations and the UExP-FNN-U fCO$_2$ (sw) showed good performance (Figure 4) similar to the results for the global scale in Ford et al. (2024a) (weighted bias = -0.18 µatm, RMSD = 20.65, N = 18226 monthly 1 degree regions). For anticyclonic eddies, we observed a smaller weighted RMSD (precision) of 19.15 µatm (N=2082 monthly matches; Figure 4a). For cyclonic eddies we observed a lower RMSD of 16.49 µatm (N = 1376; Figure 4d). Both eddy types showed small weighted biases (accuracy) and therefore we consider the UExP-FNN-U generated fCO$_2$ (sw) within eddies to sufficiently represent the eddy fCO$_2$ (sw). The differences between the within-eddy UExP-FNN-U fCO$_2$ (sw) and in situ SOCAT observations did not indicate regional biases, but did show a spatial weighting to the Northern Hemisphere where more in situ fCO$_2$ (sw) are made (Bakker et al., 2016; Figure 4c,f).

Seasonally separating the collocated within eddy in situ observations shows that the UExP-FNN-U tended to show a small weighted bias (accuracy) and smaller RMSD (precision) during winter and autumn (Figure 5a,b,g,h) compared to spring and summer (Figure 5c,d,e,f). Although winter and autumn tended to have lower collocations between in situ SOCAT observations and the UExP-FNN-U fCO$_2$ (sw) (Figure 5). These seasonal comparisons further strengthen the accuracy and precision of the UExP-FNN-U fCO$_2$ (sw) and indicates no large seasonal biases. Figure 4b and Figure 4e show that the uncertainties calculated for the fCO$_2$ (sw) were able to sufficiently represent the differences to the SOCAT observations. Thereby providing validity to the fCO$_2$ (sw) contribution to the air-sea CO$_2$ flux uncertainty budgets.

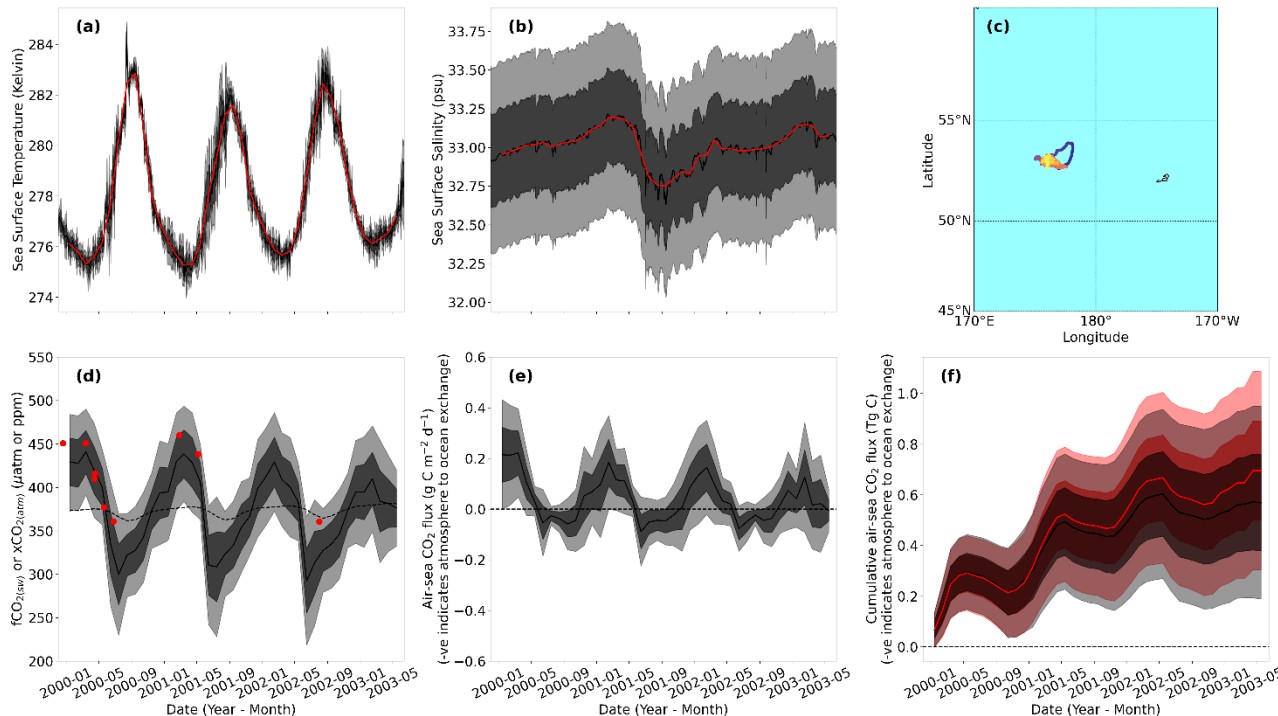

**Figure 3: Exemplar eddy trajectory (eddy 194465) in the North Pacific Ocean with calculated air-sea CO₂ fluxes (a) Sea surface temperature (SST) for the example eddy's lifetime. Black line is the daily SST, where dark grey and light grey shading indicates the 1 sigma (~67% confidence) and 2 sigma (~95% confidence) uncertainties. Red line is the median monthly SST. (b) same as (a) for sea surface salinity. (c) Geographical eddy trajectory, where colour indicates the age of eddy (blue is eddy formation and yellow is eddy dissipation). (d) Monthly timeseries of fugacity of CO₂ in seawater (fCO₂ (sw); solid line) and dry mixing ratio of CO₂ in the atmosphere (xCO₂ (atm); dashed line) for the eddy. Dark grey and light grey shading indicates the 1 sigma (~67% confidence) and 2 sigma (~95% confidence) uncertainties on the fCO₂ (sw). Red dots indicate fCO₂ (sw) in situ observations from the Surface Ocean CO₂ Atlas within the eddy. (e) same as (d) but for the air-sea CO₂ flux where a positive flux means CO₂ outgassing. Dashed black line indicates an air-sea CO₂ flux of 0. (f) same as (d) but for the cumulative air-sea CO₂ flux. Red line and banding indicate the cumulative air-sea CO₂ flux for the surrounding environment.**

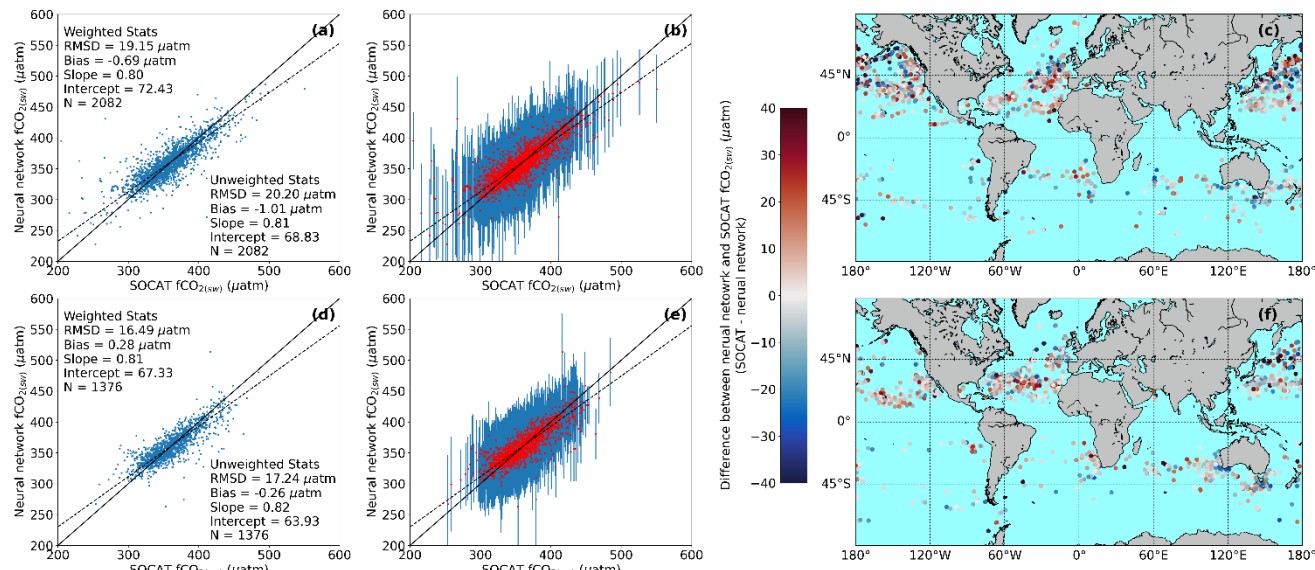

**Figure 4: (a) Comparison of the UExP-FNN-U fCO$_{2\ (sw)}$ to in situ SOCAT observations within anticyclonic eddies. Solid black line is the 1:1. Dashed line is the Type II linear regression. In text statistics are root mean square difference (RMSD), bias, slope and intercept of a Type II linear regression and number of matches (N). (b) same as (a) but showing the uncertainty on the fCO$_{2\ (sw)}$ (2 sigma; 95% confidence) as errorbars for anticyclonic eddies. (c) Difference between UExP-FNN-U fCO$_{2\ (sw)}$ to in situ SOCAT observations within anticyclonic eddies plotted as spatial residuals. (d, e and f) same as (a, b and c) but for cyclonic eddies.**


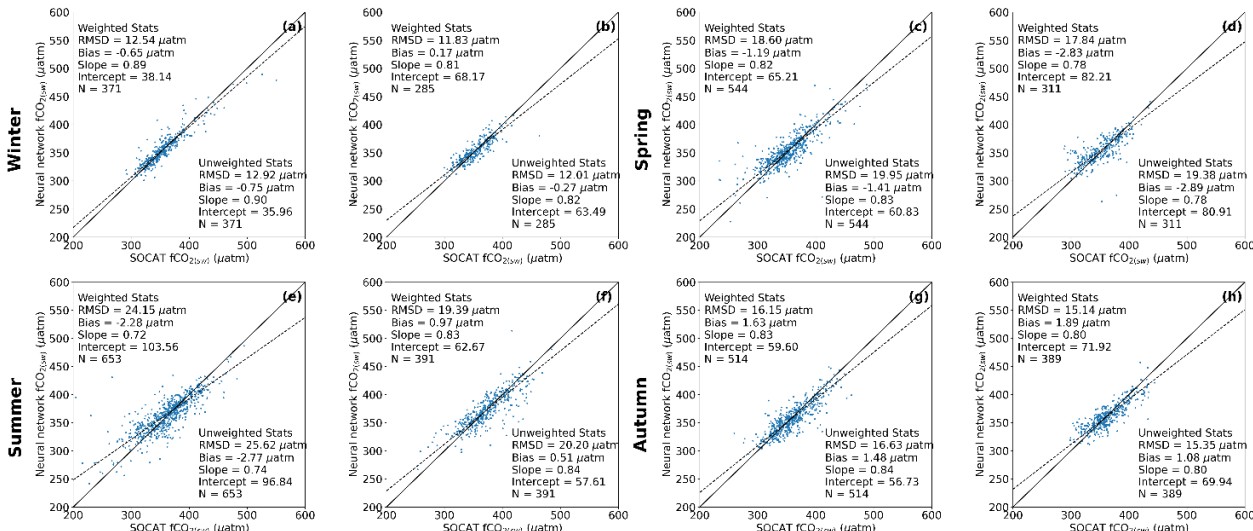

**Figure 5: (a)** Comparison of the UExP-FNN-U fCO$_{2\,(sw)}$ to in situ SOCAT observations within anticyclonic eddies during winter. Solid black line is the 1:1. Dashed line is the Type II linear regression. In text statistics are root mean square difference (RMSD), bias, slope and intercept of a Type II linear regression and number of matches (N). **(b)** same as (a) but for cyclonic eddies in the winter. **(c)** and **(d)** same as (a) and (b) for spring. **(e)** and **(f)** same as (a) and (b) for summer. **(g)** and **(h)** same as (a) and (b) for autumn.

## 3.4 Uncertainty in the mesoscale eddy cumulative air-sea CO₂ flux

Two exemplar eddies, eddy A with a lifetime of 12 months and eddy B with a lifetime of 42 months, are shown in Figure 6. These were selected to highlight the differences in the relative and absolute contributions of each uncertainty component to the total uncertainty, and how these can change over time for eddies of differing lifetimes. The absolute uncertainty magnitudes for eddy B were larger than eddy A (Figure 6b, d), but the relative contributions of each component showed similarities.

For both eddies at the end of their life, the fCO$_{2\,(sw)}$ component was the dominant source to the uncertainty for the whole lifetime, followed by the gas transfer parameterisation uncertainty. For eddy A, wind speed was the next largest contributor to the uncertainties, whereas for the eddy B, the solubility component uncertainties were larger than the wind speed uncertainty.

Throughout both eddy lifetimes the dominant uncertainty contributions changed. For eddy A, at formation showed that the 315 wind speed and solubility components were larger contributors than the gas transfer uncertainty until four months after

formation (Figure 6b). Within eddy B, the wind speed was a larger contributor than the solubility components until 12 months after formation, at which time the solubility component becomes a larger contributor (Figure 6d). Uncertainties due to the Schmidt number and $fCO_{2\,(atm)}$ terms were a small contribution to the uncertainty in both eddies.

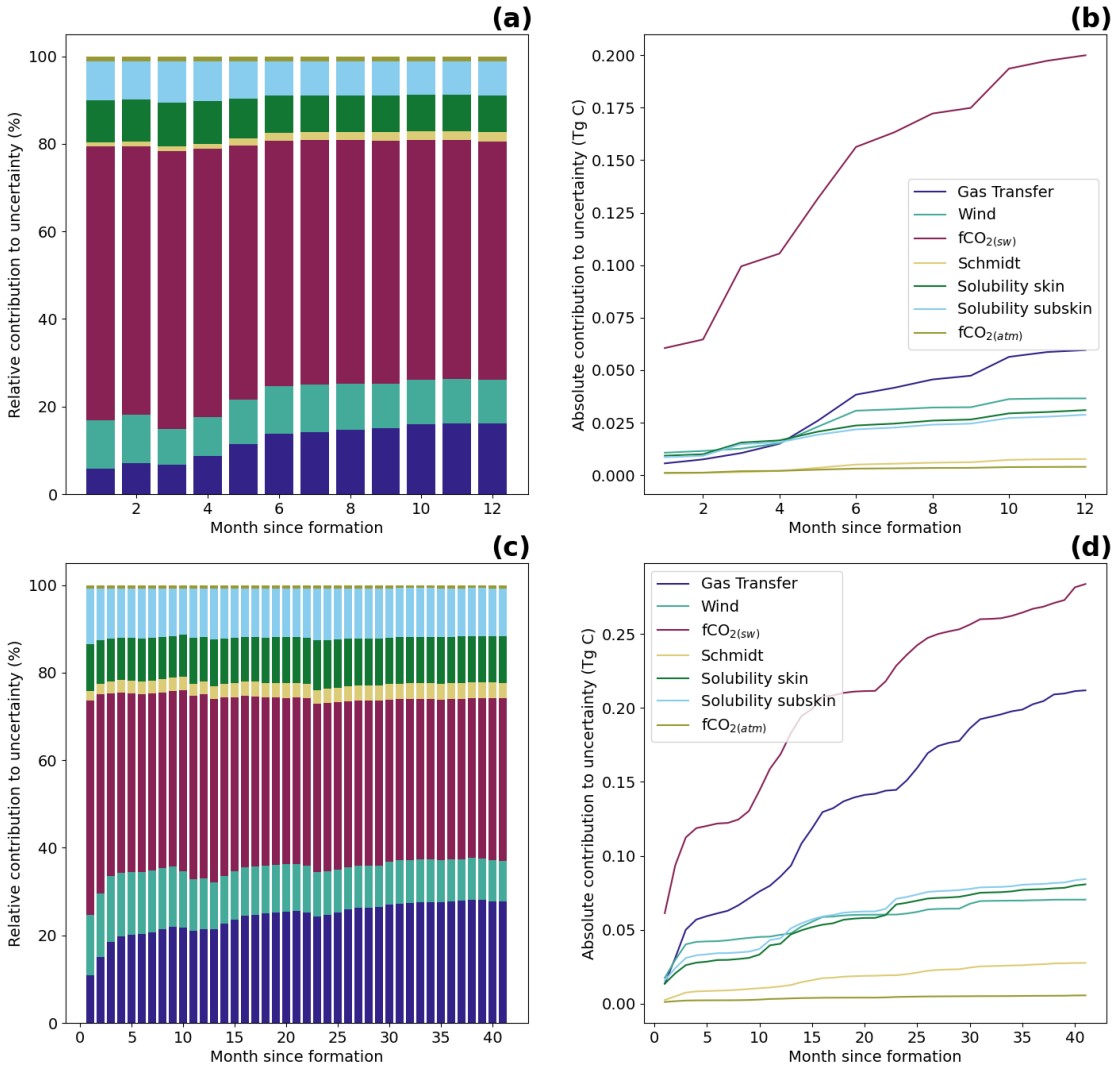

 **Figure 6: (a) The total cumulative air-sea $CO_2$ flux uncertainty (2 sigma) for an exemplar anticyclonic eddy, eddy A, (1 year lifetime; eddy 496) split into the relative contributions for the individual components. (b) The total air-sea $CO_2$ flux uncertainty in absolute terms. Legend in (b) corresponds to colours in (a). (c) and (d) same as (a) and (b) but for eddy B, an anticyclonic eddy (42 months lifetime; eddy 194465). Note different x-axis limits for (a) and (b) compared to (c) and (d).**

**3.5 Global and regional mesoscale modifications of the air-sea CO₂ flux**

An example application of the dataset was to assess the modification of the cumulative air-sea $CO_2$ flux by individual eddies at their dissipation. The analysis indicated that individual eddies could enhance (negative percentage changes) or suppress (positive percentage changes) the $CO_2$ sink (Figure 7). Both anticyclonic (Figure 7a) and cyclonic eddies (Figure 7b) showed individual eddies that were either enhancing or suppressing the air-sea $CO_2$ flux. Regional signatures in the air-sea $CO_2$ flux modification were apparent, for example anticyclonic eddies in the South Pacific and Southern Ocean had a greater tendency

to enhance the $CO_2$ sink, whereas in the Indian Ocean there was not a discernible tendency. Cyclonic eddies in the Southern Ocean indicated a larger suppression of the $CO_2$ sink than for example the North Pacific.

Considering all the eddies studied and the calculated uncertainties, anticyclonic eddies were identified to enhance the cumulative $CO_2$ flux, where these eddies acted as stronger $CO_2$ sink (weaker $CO_2$ source) by 4.5 ± 2.8 % (95 % confidence). Cyclonic eddies indicated a slight suppression of the cumulative air-sea $CO_2$ flux by 0.7 ± 2.6 %, acting overall to weaken

the $CO_2$ sinks (or as stronger $CO_2$ sources). Here we note, at the 95 % confidence the cumulative $CO_2$ sink enhancement by anticyclonic eddies was significantly difference from 0 (i.e the confidence interval did not include 0) when uncertainties were accounted for, but this was not significantly different from 0 for cyclonic eddies.

The regional differences can be emphasised by considering median eddy modifications within different regional subsets (Figure 8) instead of globally (Figure 8c, d). The eddy modification of $CO_2$ fluxes within the regions showed differing

magnitudes that fall within different significance bands when the uncertainties are accounted for. For example, the Southern Ocean shows an anticyclonic enhancement of the $CO_2$ sink of 5.7 ± 5.0 % (significant at 95 % confidence), with cyclonic eddies suppressing the $CO_2$ sink by 2.5 ± 4.6 %. In the North Pacific, we find similar results where anticyclonic eddies enhance by 5.6 ± 5.2 %, and cyclonic eddies suppress by 1.7 ± 7.4 %. Consistent results were found for the South Pacific but noting the cyclonic eddies showed a larger uncertainty interval of 11.5 %. The South Atlantic Ocean showed the anticyclonic

enhancement of the sink by 0.3 ± 15.0 % and cyclonic eddies appear to enhance the $CO_2$ sink by 0.7 ± 13.7 %. The uncertainty intervals on these are however the largest of any region, likely due to the lowest number of eddies considered.

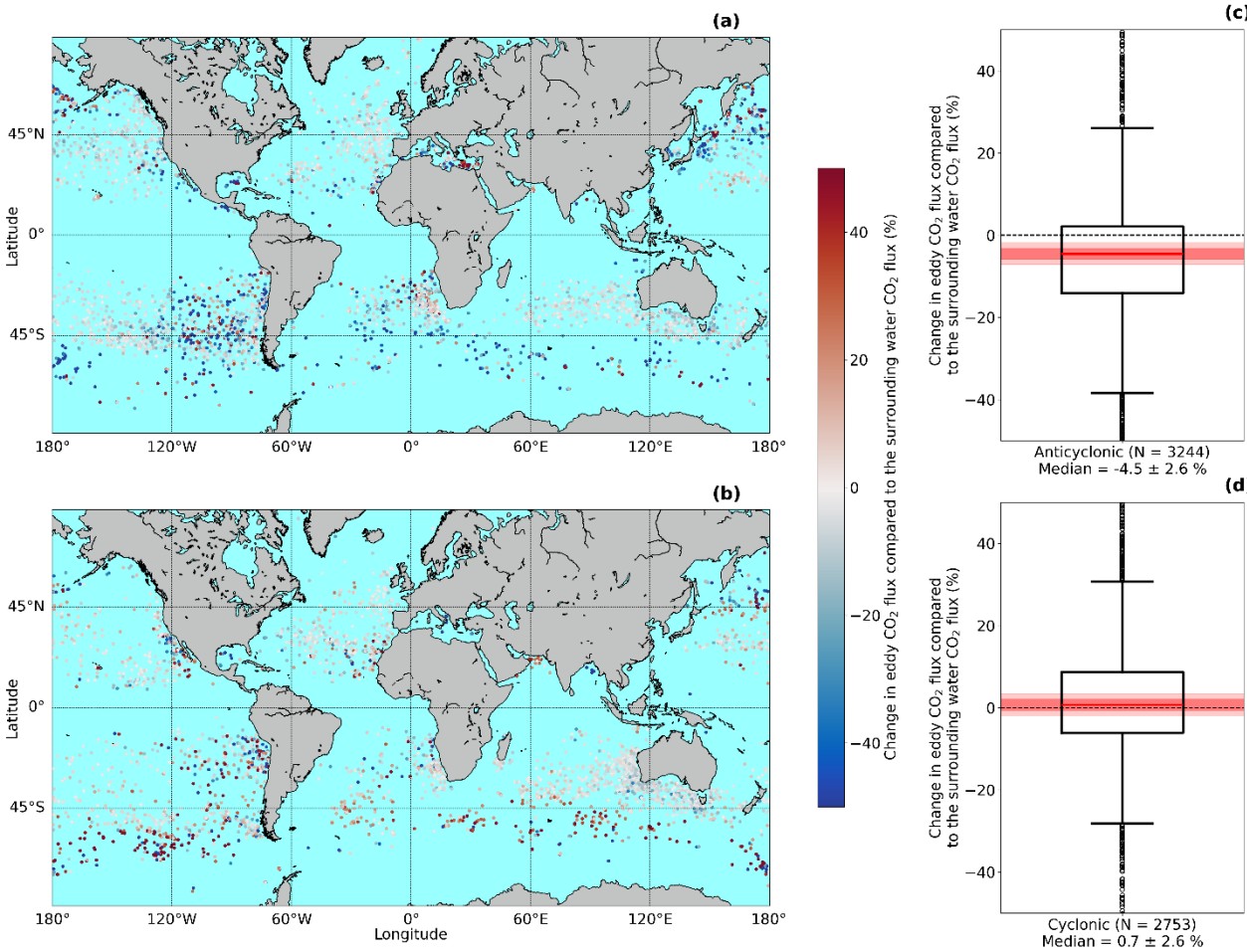

**Figure 7: (a) Geographical distribution of the anticyclonic eddies' modification of the cumulative air-sea CO₂ flux. Negative values indicate a stronger CO₂ sink (weaker CO₂ source), and positive values indicate a weaker CO₂ sink (stronger CO₂ source). (b) same as (a) for the cyclonic eddies. (c) Box plot showing the anticyclonic eddy modification of the air-sea CO₂ flux. Red line indicates the median, box indicates the 25th and 75th quartiles, whiskers extend from the 25ᵗʰ and 75ᵗʰ quartiles by 1.5 interquartile ranges. Circles indicate data considered outliers. Dark red shading indicates the 1 sigma (~68% confidence) uncertainty on the median by propagating the air-sea CO₂ flux uncertainties using a Monte Carlo uncertainty propagation. Light red shading indicates the 2 sigma uncertainty on the median (~95% confidence). X-axis label shows number of eddies (N), the median modification with the 2 sigma uncertainty. (d) same as (c) but for the cyclonic eddies. Basemap in (a) and (b) from Natural Earth v4.0.0 (https://www.naturalearthdata.com/).**

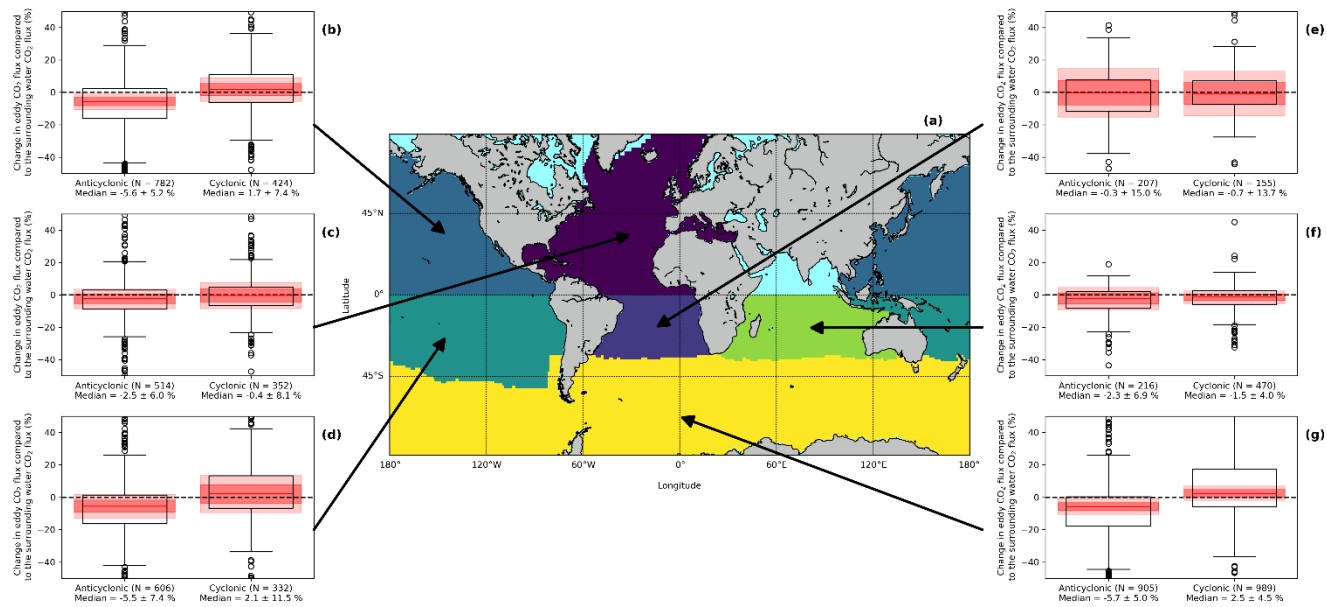

**Figure 8: (a)** Ocean basins considered for further analysis, with a colour for each region. Regions follow the RECCAP2 ocean basin definition, but each basin was split at the Equator into North and South. North Indian Ocean was removed due to low number of eddies analysed. **(b)** Box plot showing the eddy modification of the cumulative air-sea $CO_2$ flux for the region shown with the arrow. Red line indicates the median, box indicates the 25th and 75th quartiles, whiskers extend from the $25^{th}$ and $75^{th}$ quartiles by 1.5 interquartile ranges. Circles indicate data considered outliers (greater than 1.5 interquartile ranges outside the $25^{th}$ and $75^{th}$ percentile). Dark red shading indicates the 1 sigma (~68% confidence) uncertainty on the median by propagating the air-sea $CO_2$ flux uncertainties using a Monte Carlo uncertainty propagation. Light red shading indicates the 2 sigma uncertainty on the median (~95% confidence). X-axis label shows number of eddy (N), the median modification with the 2 sigma uncertainty. **(c), (d), (e), (f), (g)** same as (b) for their respective regions identified by the arrow. Basemap in (a) from Natural Earth v4.0.0 (**https://www.naturalearthdata.com/**).

## 4. Discussion

### 4.1 Mesoscale eddy air-sea CO₂ fluxes and uncertainties

The mesoscale eddy air-sea $CO_2$ fluxes provide both the $CO_2$ fluxes for each month with uncertainties and the corresponding environmental data (i.e SST, SSS) within and outside of each eddy (Figure 3). These data allow a range of analyses to be conducted, for example, in this study, we show how the mesoscale modification of the air-sea $CO_2$ flux can be determined from these data regionally (Figure 7; Figure 8) or could be evaluated through time (e.g. Table S1 provides global decadal median mesoscale modifications suggesting an increasing enhancement of the $CO_2$ sink). Other potential applications could include, analysing the thermal and non-thermal components in driving the global eddy modified air-sea $CO_2$ fluxes (as illustrated by Ford et al. (2023) for the South Atlantic), or for investigating nutrient entrainment within the eddies and how it links to biological variations within the eddy track, or the variability in phytoplankton biomass and / or productivity within

the eddies which are important for improving our understanding of carbon rate dynamics, and their impacts on ecology and biodiversity. The dataset presented here therefore provides the basis for a wide range of studies to assess the evolution of

mesoscale eddies and their air-sea $CO_2$ fluxes alongside understanding the linkages with their localised environmental conditions.

The dataset air-sea $CO_2$ flux estimates are accompanied by a comprehensive uncertainty budget developed by Ford et al. (2024a) (Figure 3; Figure 6). This is the first dataset of eddy air-sea $CO_2$ fluxes to include a uncertainty budget that has been built on the principles where all known sources of uncertainty are systematically considered (however small) and propagated

to the final uncertainty using standard propagation techniques and a well-established uncertainty framework (BIPM, 2008; Taylor, 1997). The budget therefore provides an uncertainty on each air-sea $CO_2$ flux estimate, and the $fCO_{2\,(sw)}$, which can be accounted for within further analyses (e.g. as used in Ford et al., 2021, 2022b) and aids in assigning confidence to any results, as demonstrated in the example results that have been presented.

The comparisons between the UExP-FNN-U $fCO_{2\,(sw)}$ and SOCAT $fCO_{2\,(sw)}$ observations within eddies provide further

confidence in the retrieved UExP-FNN-U $fCO_{2\,(sw)}$ and resulting air-sea $CO_2$ fluxes (Figure 4; Figure 5). We showed that for both the anticyclonic and cyclonic eddies the within eddy accuracy (bias) and precision (RMSD) showed greater performance when compared to the global scale performance of these approaches (Ford et al., 2024a). This result was consistent with Ford et al. (2023) for the South Atlantic Ocean, who showed that both eddy types were well represented by the neural network approach (except Ford et al. (2023) determined this from a lower number of crossover data points than

presented here). Li et al. (2025) also showed for their neural network approach, similar accuracy and precision results for the $fCO_{2\,(sw)}$ within eddies for four western boundary current regions. Although, we did observe a slightly lower precision during the spring and summer, which could be due to the lack of a biological predictor (e.g chl-a) reducing the ability of the UExP-FNN-U to capture these dynamics (Ford et al., 2022a) (Figure 5).  These results also provide validity to the calculated $fCO_2$ $_{(sw)}$ uncertainties, which in the majority of cases are dominated by the $fCO_{2\,(sw)}$ evaluation uncertainty component. As the

retrieved within eddy $fCO_2$ $_{(sw)}$ bias and RMSD showed greater performance compared the global UExP-FNN-U performance (given in Ford et al., 2024) we are confident in the UExP-FNN-U $fCO_{2\,(sw)}$ and uncertainty estimates within the eddies.

Within the UEx-L-Eddies we provide a secondary $fCO_{2\,(sw)}$ estimate (and associated air-sea $CO_2$ fluxes) from a global $fCO_2$ $_{(sw)}$ neural network, which included chl-a as a predictor. We include the additional neural network because Ford et al. (2022a)

highlighted that the inclusion of more representative biological parameters improved the regional estimation of $fCO_{2\,(sw)}$ in the South Atlantic Ocean, which is likely to be the same for other regions. Previous studies have shown the importance of biological modulation of $fCO_{2\,(sw)}$ within eddies (Orselli et al., 2019b), the resulting $CO_2$ fluxes, and how the importance changes over the eddy lifetime (Ford et al., 2023). This additional neural network showed similar but slightly improved precision (lower weighted RMSD) when compared to the in situ SOCAT observations, although to a lower number of data

points (Figure S2; anticyclonic bias = -0.92 uatm, RMSD = 17.05 μatm, N = 1914; cyclonic bias = 0.05 μatm, RMSD = 14.31 μatm, N = 1272). In addition, the seasonal breakdown of the comparisons between the within eddy UExP-FNN-U with chl-a $fCO_{2 (sw)}$ and the in situ $fCO_{2 (sw)}$ showed an increase in the performance of this neural network during spring and summer, highlighting the improvements from chl-a being added as a predictor (Figure S3). These estimates are however restricted to regions between 50 ºN and 50 ºS due to the availability of ocean colour chl-a data in polar winter (i.e for a full eddy timeseries the eddy must remain within the available ocean colour data).

The impact on the modification of the cumulative air-sea $CO_2$ flux by mesoscale eddies due to including chl-a within the UExP-FNN-U can be assessed by replicating Figure 8, but using the secondary $fCO_{2 (sw)}$ and resulting air-sea $CO_2$ fluxes (Figure 9). Figure 9 shows the regional modification of the air-sea $CO_2$ fluxes by eddies where both neural network variants are able to estimate the $fCO_{2 (sw)}$ (i.e we show a subset of the eddies in Figure 8). In all regions both neural networks retrieve a similar signature, but the chl-a version generally suggests a stronger enhancement (or weaker suppression) of the $CO_2$ sink compared to the UExP-FNN-U without chl-a. Notably the South Pacific Ocean and Southern Ocean show larger differences although in all cases these differences fall within the uncertainties. We therefore provide the secondary neural network to further aid in understanding the processes that are driving mesoscale eddy modification of the air-sea $CO_2$ fluxes.


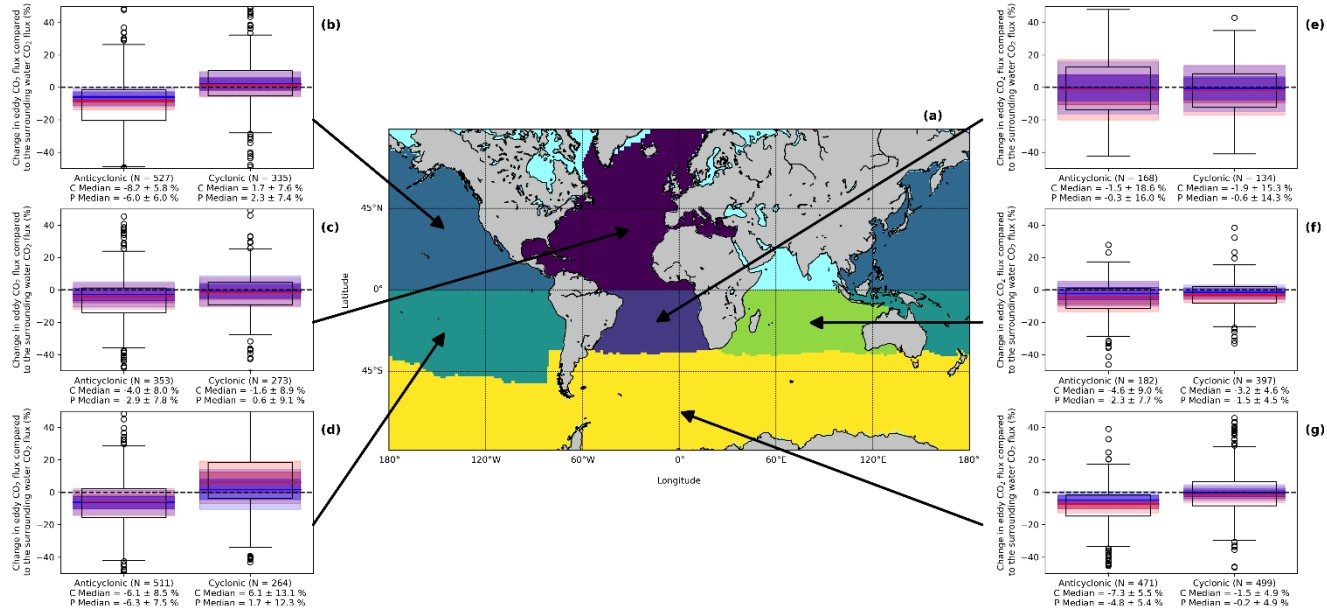

**Figure 9: (a) Ocean basins considered for further analysis, with a colour for each region. Regions follow the RECCAP2 ocean basin definition, but each basin was split at the Equator into North and South. North Indian Ocean was removed due to low number of eddies analysed. (b) Box plot showing the eddy modification of the air-sea CO₂ flux using the chl-a version of the UExP-FNN-U for the region shown with the arrow. Red line indicates the median, box indicates the 25th and 75th quartiles, whiskers extend from the 25ᵗʰ and 75ᵗʰ quartiles by 1.5 interquartile ranges. Circles indicate data considered outliers (greater than 1.5 interquartile ranges outside the 25ᵗʰ and 75ᵗʰ percentile). Dark red shading indicates the 1 sigma (~68% confidence) uncertainty on the median by propagating the air-sea CO₂ flux uncertainties using a Monte Carlo uncertainty propagation. Light red shading indicates the 2 sigma uncertainty on the median (~95% confidence). Blue line and shading indicates the same but for the UExP-FNN-U without chl-a. X-axis label shows number of eddy (N), the median modification with the 2 sigma uncertainty for the chl-a version of the UExP-FNN-U labelled with a C, and the UExP-FNN-U without chl-a labelled with a P. (c), (d), (e), (f), (g) same as (b) for their respective regions identified by the arrow. Basemap in (a) from Natural Earth v4.0.0 (https://www.naturalearthdata.com/).**

Previous eddy trajectory datasets have been produced, for example Dong et al. (2022a), which include environmental datasets (e.g. SST) that can be used to understand the effects of eddies on physical and biological properties. The UEx-L-Eddies however extends the principles of these datasets to include air-sea $CO_2$ fluxes but also has a focus on climate quality dataset (i.e the ESA CCI datasets) and provides comprehensive uncertainties. Therefore it provides a robust dataset for understanding long-lived eddy effects on the surface properties and air-sea $CO_2$ fluxes. In the future, we plan to include in situ observations by Biogeochemical Argo floats (BGC-Argo; Roemmich et al., 2019), which could be used to provide in situ based $fCO_{2\,(sw)}$ and air-sea $CO_2$ fluxes to further verify the air-sea $CO_2$ fluxes (e.g., as suggested by Keppler et al. (2024)).

**4.2 Comparison to previous global and regional eddy modifications of the air-sea CO₂ fluxes**

Previous studies have investigated the effect of mesoscale eddies on global and regional air-sea $CO_2$ fluxes (Table 2). Guo and Timmermans (2024) evaluate the cumulative effect of mesoscale variability on the air-sea $CO_2$ flux globally, which they find enhances the global air-sea $CO_2$ flux by 0.72 Mt C yr$^{-1}$, or 0.72 Tg C yr$^{-1}$. With the UEx-L-Eddies, if the individual eddy air-sea $CO_2$ flux modifications are summed for the whole dataset, we find a global cumulative enhancement of the ocean $CO_2$ sink by long-lived mesoscale eddies of 75 ± 33 Tg C between 1993 and 2022. This would be equivalent to 2.7 ± 1.1 Tg C yr$^{-1}$ (95 % confidence interval). The calculated uncertainties with the UEx-L-Eddies allows robust uncertainty estimates to be provided alongside further analyses of the individual eddies, allowing significance of comparisons to be assessed. Differences here may be due to Guo and Timmermans (2024) including mesoscale variability not associated with mesoscale eddies (such as filaments, and current meanders), as their method does not track individual eddies. It could also be due to the UEx-L-Eddies only covering long-lived eddies, that represent 0.4 % of eddies within the META3.2 trajectories dataset and therefore misses the contribution of smaller eddies (Pegliasco et al., 2022b) that would be included with Guo and Timmermans (2024).

Li et al. (2025) showed for the Kuroshio current that anticyclonic eddies enhanced the $CO_2$ sink by 15 ± 1.73 %, and cyclonic eddies reduced the $CO_2$ sink by 5.7 ± 1.5 %. Similar results were also shown for the Gulf Stream. Both the Gulf Stream and the Kuroshio current are dominated by short-lived eddies (e.g., those that survive for less than 1 year) in comparison to the long-lived eddies studied within the UEx-L-Eddies dataset, and therefore comparing these two estimates is inappropriate. However, our regional results for the North Pacific and North Atlantic Oceans do show a consistent direction of change (i.e., an enhanced sink) but with smaller magnitudes (Figure 8).

Keppler et al. (2024) investigate the role of mesoscale eddies in modifying the air-sea $CO_2$ flux in the Southern Ocean using Biogeochemical Argo profilers between April 2014 to December 2022. They find anticyclonic eddies enhanced the air-sea $CO_2$ sink by 7 ± 2 % and cyclonic eddies reduced the air-sea $CO_2$ flux by 2 ± 2 % (1 sigma uncertainties). Within the UEx-L-Eddies, we found that anticyclonic eddies enhanced the $CO_2$ sink by 5.7 ± 5.0 % (2 sigma uncertainties), and cyclonic eddies reduced the sink by 2.5 ± 4.5 % between 1993 and 2022 (Figure 8g). These consistent results provide confidence to the air-sea $CO_2$ flux estimates within the UEx-L-Eddies.

Ford et al. (2023) showed that within the South Atlantic Ocean, anticyclonic (N = 36) and cyclonic (N = 31) eddies enhanced the $CO_2$ sink by 3.7 % and 1.7 %, respectively. In our analysis for the South Atlantic Ocean (Figure 8e) we showed that anticyclonic enhanced the $CO_2$ sink by 0.3 ± 15.0 (N = 207) and cyclonic eddies enhanced the $CO_2$ sink by 0.7 ± 13.7 % (N = 155) respectively, where confidence intervals are expressed as 95 % confidence. Within this dataset, we consider ~5 times more eddies than Ford et al. (2023) and find that the air-sea $CO_2$ flux uncertainties have a large effect on our resulting confidence, making the results indistinguishable at 95 % confidence (even at 67 % confidence the two are indistinguishable).

The comparison highlights the importance of the calculated uncertainties and their use within further analyses and comparisons with other air-sea $CO_2$ fluxes.

The UEx-L-Eddies identifies differences in the mesoscale eddy modification of the cumulative air-sea $CO_2$ flux between anticyclonic and cyclonic eddies globally and regionally consistent with previous analyses. The driving mechanisms for these differences have been investigated in previous work. For example, Li et al. (2025) suggest that the competing changes in dissolved inorganic carbon and biological processes through eddy pumping contribute to the observed mesoscale eddy modification of the air-sea $CO_2$ flux. Additionally, Keppler et al. (2024) showed that the mesoscale modification of the air-

sea $CO_2$ flux had significant seasonal variability in the Southern Ocean, indicating that underlying driving processes could vary throughout the individual eddies lifetime. Ford et al. (2023) showed that the changes in air-sea $CO_2$ flux in mesoscale eddies could be attributed to changes in the competing biological and physical processes. Although a comprehensive analysis of the driving mechanism is beyond the scope of this manuscript, the UEx-L-Eddies shows regional (Figure 8) and seasonal variability in the mesoscale eddy modification of the air-sea $CO_2$ flux (e.g. Figure S4 shows anticyclonic eddies have

stronger uptake in winter). The underlying environmental parameters (e.g. SST, MLD) could therefore be used to investigate the driving mechanisms for these differences in the mesoscale modification.

**Table 2: Summary of methodologies in previous studies used to estimate the eddy modification of the air-sea CO₂ flux. pCO₂ (sw) is the partial pressure of CO₂ in seawater.**

| | This study | Guo and Timmermans (2024) | Li et al. (2025) | Keppler et al. (2024) | Ford et al. (2023) |
|---|---|---|---|---|---|
| Eddy Dataset (or decomposition approach) | META 3.2 | Mesoscale signature decomposition | META 3.2 | META 3.2 | META 3.1exp |
| Lifetimes considered | > 1 year | N/A | >12 weeks | >=10 days | >1 year |
| Radius Threshold | No criteria | N/A | No criteria | >40km | No criteria |
| $fCO_{2\,(sw)}$ estimation method | Global $fCO_{2\,(sw)}$ neural network approach | Eddy resolving model | Regional $pCO_{2\,(sw)}$ neural network approach | In situ pH with neural network Total Alkalinity | Regional $pCO_{2\,(sw)}$ neural network-approach |
| Temporal Coverage | January 1993 to December 2022 | 1982 to 2000 | July 2002 to 1 January 2022 | April 2014 to February 2022 | July 2002 to December 2018 |
| Spatial Domain | Global | Global | Western Boundary Current (Kuroshio and Gulf Stream) | Southern Ocean | South Atlantic Ocean |
| Air sea CO₂ flux uncertainty treatment | Comprehensive uncertainty | N/A | $fCO_{2\,(sw)}$ and gas transfer considered | Standard error of observations | $fCO_{2\,(sw)}$ and gas transfer considered |


### 4.3 Limitations when using the UEx-L-Eddies

For some eddies the daily environmental data can have missing values even for complete coverage data (for example, the CCI-SST). These gaps stem from the META3.2 eddy trajectories dataset where the polygon to define the limits of the eddy

does not form correctly, and therefore we were unable to extract values where the polygon was undefined. No exclusion or interpolation mechanism was implemented as these data gaps affect a mean of 2 % (maximum = 15 %) of an individual eddy daily timeseries, which occur randomly through the timeseries, and therefore the impact on the monthly median statistics are minimal.

The UEx-L-Eddies dataset focusses on larger, long-lived eddies (lifetimes greater than a year). This criteria will regionally exclude eddies within, for example, highly dynamic western boundary currents where shorter lived eddies often dominate (Figure 2c, d). Smith et al. (2023) however show that eddies with smaller radii generally have the same anomaly direction but with weaker magnitudes when compared to larger eddies. A previous study (Pegliasco et al., 2022b) identified that the shorter lived eddies within the Mesoscale Eddy Product (the same product used within this study) generally have smaller radii then the longer lived eddies. Therefore we would expect similar anomalies but of smaller magnitude when studying shorter lived eddies.

## 5. Summary

The UEx-L-Eddies is a dataset of the air-sea $CO_2$ fluxes for (N=5996) long lived mesoscale eddies calculated in a Lagrangian mode within the global ocean. We use a global $fCO_{2\ (sw)}$ neural network (as used within one dataset submitted to the Global Carbon Budget called UExP-FNN-U) to estimate the $fCO_{2\ (sw)}$ within the eddies at a monthly resolution. We prioritise the use of climate quality datasets within the analysis. The air-sea $CO_2$ fluxes (also calculated following the methods of UExP-FNN-U) are accompanied by a comprehensive uncertainty budget (using a published methodology), that considers all known sources of uncertainty. We show for an exemplar eddy that the seasonal cycles of the eddy $fCO_{2\ (sw)}$ and air-sea $CO_2$ fluxes are captured and can be cumulatively added to assess the $CO_2$ uptake (or outgassing) of individual eddies. The comprehensive air-sea $CO_2$ flux uncertainties provide a robust basis for assessing the confidence in the eddy air-sea $CO_2$ flux estimates and can be propagated to further analysis. This illustrates how the importance of the different uncertainty components can change through time highlighting the shortfall of only quantifying selected contributions to the uncertainties or assuming fixed values.

Within the uncertainty assessment, we find that the $fCO_{2\ (sw)}$ in the eddies are estimated with an accuracy (bias) of -0.69 µatm and a precision (RMSD) of 19.15 µatm for anticyclonic (N = 2082), and accuracy of 0.28 µatm and a precision of 16.49 µatm for cyclonic eddies (N = 1376). These accuracy and precision estimates provide validity to the neural network $fCO_{2\ (sw)}$. We demonstrate a use case of the UEx-L-Eddies dataset to evaluate the air-sea $CO_2$ flux modification, and resultant integrated net $CO_2$ sink, by long-lived mesoscale eddies, globally and regionally. We find that anticyclonic eddies enhance the net sink by 4.5 ± 2.8 % (N = 3244), and cyclonic eddies suppress by 0.7 ± 2.6 % (N = 2752) where uncertainties are the 95% confidence interval. Regional differences in the eddy modification are observed, for example within the Southern

Ocean, anticyclonic eddies enhanced the $CO_2$ sink by 5.7 ± 5.0 %, and cyclonic eddies reduced the sink by 2.5 ± 4.5 %. We demonstrate how the use case results are consistent with previous regional analyses. Our example also highlighted the importance of using the accompanying uncertainty information when comparing studies, and caution should be taken in drawing conclusions from small samples or individual eddies, without considering the underlying comprehensive uncertainty budgets for the air-sea $CO_2$ fluxes. The data presented could now be used to understand the processes occurring within these

eddies that are driving these modifications of the air-sea $CO_2$ fluxes, and how regionally these processes may vary.

**Author Contributions**

DJF, GHT, VK, KS and JDS conceived the study and the methodology. DJF performed the analysis, testing of the dataset and wrote the original draft. All authors provided input to the final manuscript.

**Competing Interests**

The authors declare no competing interests.

**Data and Code Availability**

The code for the analysis is available, and version controlled on Github at https://github.com/JamieLab/pyEddyCO2. The UEx-L-Eddies dataset are available on Zenodo (https://doi.org/10.5281/ZENODO.16355763; Ford et al., 2025). The AVISO+ eddies trajectories data (META 3.2) was retrieved from AVISO+ (https://doi.org/10.24400/527896/A01-

2022.005.220209; Pegliasco et al., 2022a). The CCI-SST climate record (v3.0) were retrieved from CEDA (https://doi.org/10.5285/4A9654136A7148E39B7FEB56F8BB02D2; Good and Embury, 2024). The OC-CCI chl-a (v6) were retrieved from CEDA (https://doi.org/10.5285/5011D22AAE5A4671B0CBC7D05C56C4F0; Sathyendranath et al., 2023). The CMEMS GLORYS12V1 SSS and MLD were retrieved from CMEMS (https://doi.org/10.48670/moi-00021; CMEMS, 2021). The CCMP wind speeds (v3.1) were retrieved from Remote Sensing Systems (https://doi.org/10.56236/rss-

uv6h30; Remote Sensing Systems et al., 2022). The $xCO_2$ (atm) were retrieved from NOAA-GML (https://doi.org/10.15138/DVNP-F961; Lan et al., 2023). In situ SOCAT observations that have been recalculated to a consistent depth and temperature dataset were retrieved from Zenodo (https://doi.org/10.5281/zenodo.15706025; Ford et al., 2024d).

**Acknowledgements**

DJF and JDS were supported by funding from the European Space Agency under the projects 'Satellite-based observations of Carbon in the Ocean: Pools, Fluxes and Exchanges' (SCOPE; 4000142532/23/I-DT) and 'Ocean Carbon for Climate' (OC4C; 3-18399/24/I-NB). GHT and VK were supported by The Atlantic Meridional Transect is funded by the UK Natural Environment Research Council through its National Capability Long-term Single Centre Science Programme, Atlantic Climate and Environment Strategic Science - AtlantiS (grant number NE/Y005589/1). This study contributes to the

international IMBeR project and is contribution number 423 of the AMT programme.

The Surface Ocean $CO_2$ Atlas (SOCAT) is an international effort, endorsed by the International Ocean Carbon Coordination Project (IOCCP), the Surface Ocean Lower Atmosphere Study (SOLAS) and the Integrated Marine Biosphere Research (IMBeR) program, to deliver a uniformly quality-controlled surface ocean $CO_2$ database. The many researchers and funding

agencies responsible for the collection of data and quality control are thanked for their contributions to SOCAT. For the purpose of open access, the authors have applied a Creative Commons Attribution (CC BY) licence to any Author Accepted Manuscript version arising from this submission.

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
