# Peer review of "UEx-L-Eddies: Decadal and global long-lived mesoscale eddy trajectories with coincident air-sea CO2 fluxes and environmental conditions"

_Earth System Science Data, 2025_

## Author Comment (AC1)

Dear Editor and Reviewers,

We thank you for the reviews of our manuscript. We greatly appreciate receiving these detailed and constructive reviews and they have helped us to improve the paper considerably. Line numbers within this document refer to the tracked change version of the manuscript.

During the revision process we identified an error within our code which was specific to eddies that crossed the international date line. In these cases there was an error when the in situ $fCO_{2 (sw)}$ observations were matched to the eddy. We have now corrected this error which results in ~1,100 fewer matches between eddies and in situ $fCO_{2 (sw)}$ observations. This has meant that some statistics in the paper have been updated but remain robust for anticyclonic (N = 2082) and cyclonic eddies (N = 1376).

The Zenodo repository has been updated with the latest version of our dataset (v0-3).

Yours sincerely,

Daniel J. Ford
* * *
Reviewer 1

This manuscript presents a global dataset of long-lived mesoscale eddies (1993–2022) that includes coincident environmental variables, neural-network–based estimates of surface ocean fugacity of $CO_2$ ($fCO_2$(sw)), and derived air–sea $CO_2$ fluxes with comprehensive uncertainty budgets. The dataset builds on the authors' earlier regional work by integrating a satellite-derived global eddy atlas, reanalysis products, and a refined neural-network methodology (UExO-FNN-U) for estimating $fCO_2$(sw). Using this global dataset, the authors investigate how long-lived eddies modulate global air–sea $CO_2$ fluxes and compare their results with other recent estimates obtained using different methods. The findings suggest that anticyclonic eddies tend to enhance the $CO_2$ sink, while cyclonic eddies slightly reduce it, although the underlying mechanisms remain unclear. Overall, this dataset represents a valuable contribution to the community by improving our understanding of how coherent mesoscale eddies influence air–sea $CO_2$ exchange. I recommend publication after the following concerns are addressed:

***Response:*** We thank the reviewer for their appraisal of our manuscript. We have addressed all their comments below.

1. The title references both air–sea $CO_2$ fluxes and "biogeochemical conditions," but the manuscript provides limited discussion or detail on the latter. It would be useful to clarify what is meant by "biogeochemical conditions" in this context and to explicitly describe what variables are included in the dataset beyond those directly used for the flux estimates.

***Response:*** *We have now modified the title of the manuscript changing 'biogeochemical conditions' to 'environmental conditions'. As the reviewer highlights, we have not included, at this time, additional biogeochemical observations beyond those required by the air-sea $CO_2$ flux calculations. The manuscript title now reads "UEx-L-Eddies: Decadal and global long-lived mesoscale eddy trajectories with coincident air-sea $CO_2$ fluxes and environmental conditions".*

2. The rationale for restricting the analysis to "long-lived" eddies (>1 year) is not well justified. While it is plausible that large, long-lived eddies exert stronger influence on air–sea $CO_2$ exchange, this assumption should be clearly articulated. Moreover, excluding shorter-lived eddies may bias the results, as acknowledged in the comparison with previous studies. The limitations of this choice should be discussed more explicitly. In addition, Figure 1 (and the associated discussion) would benefit from showing a percentage map of long-lived eddies relative to the total number of eddies tracked in META. This would provide readers with a clearer sense of how representative the dataset is compared to the broader eddy

population. If the percentage of long-lived eddies differs substantially across regions, this may imply that eddy-induced carbon fluxes arise from different mechanisms depending on the local background dynamics (e.g., Gulf Stream vs. subtropical gyres). In addition, this choice of "long-lived" eddies excludes many eddies from the most energetic and eddy-rich regions, such as the Gulf Stream, the Kuroshio, and other western boundary currents and their extensions, where short-lived but highly dynamic eddies dominate as the authors noted in lines 390. This limitation should be discussed more explicitly.

*Response: We have now explicitly stated the reasons for our focus on long-lived eddies. Firstly, as the reviewer highlights these are the features that likely exert a larger influence on the air-sea $CO_2$ flux, and therefore the signal of the mesoscale modification will be larger against the background air-sea $CO_2$ flux dynamics (for example as shown in Smith et al., 2023). Secondly our selection is a computational choice, whereby the extensive set of shorter lived eddies would require computation time that exceeds the current available resources. This information has been added at Lines 101-105 which reads "The focus on these long-lived eddies was due to their presence likely exhibiting a larger influence on the air-sea $CO_2$ flux (e.g. Smith et al., 2023). Additionally, the selection was due to computational limitations in running the analysis for the extensive set of shorter lived eddies within the dataset. We are working to extend the analysis to shorter lived eddies but currently the focus remains on long lived eddies.".*

*We have now discussed the limitation of long lived eddies and suggested the potential future work direction of studying the mesoscale modification for the shorter lived eddies within the limitations section. The information has been added at Lines 518-524 which reads "The UEx-L-Eddies dataset focusses on larger, long-lived eddies (lifetimes greater than a year). This criteria will regionally exclude eddies within, for example, highly dynamic western boundary currents where shorter lived eddies often dominate (Figure 2c, d). Smith et al. (2023) however show that eddies with smaller radii generally have the same anomaly direction but with weaker magnitudes when compared to larger eddies. A previous study (Pegliasco et al., 2022) identified that the shorter lived eddies within the Mesoscale Eddy Product (the same product used within this study) generally have smaller radii then the longer lived eddies. Therefore we would expect similar anomalies but of smaller magnitude when studying shorter lived eddies.".*

*As suggested, we have now added subplots to Figure 2 that displays the percentage of long lived eddies that form in 1 degree regions compared to the total number of eddies that form. The updated Figure 2 is displayed below this response. The new subplots highlight that the long-lived eddies can constitute up to 30% of the total eddy formations. But as highlighted by the reviewer the percentage of long lived eddies falls to near 0% in western boundary current regions where shorter lived eddy formations are dominant.*

[Figure]

**Figure 2:** (a) The cumulative air-sea $CO_2$ flux into the anticyclonic eddies where the scatter points are plotted at the formation location of each eddy. (b) same as (a) but for cyclonic eddies. (c) The percentage of long lived anticyclonic eddy trajectories compared to all eddy trajectories that form in 1 degree by 1 degree regions. (d) same as (c) but for cyclonic eddies. Basemap from Natural Earth v4.0.0 (https://www.naturalearthdata.com/). Supplementary Figure S1 shows the equivalent of (a) and (b) in Tg C d$^{-1}$ to remove the differences in eddy lifetime.

3. The results show that long-lived anticyclonic eddies enhance the $CO_2$ sink while cyclonic eddies reduce it, but the mechanisms remain underexplored. Some discussions of potential physical and/or biogeochemical processes (e.g., temperature effects, stratification, nutrient dynamics, biological production) that could drive these differences would strengthen the manuscript and help readers interpret the findings.

*Response: We thank the reviewer for the suggestion. We have now added a paragraph that explores the potential mechanisms driving these differences which are supported by the previous research. A comprehensive analysis of the driving mechanisms is beyond the scope of this manuscript. The new paragraph at Lines 494-505 reads "The UEx-L-Eddies identifies differences in the mesoscale eddy modification of the cumulative air-sea $CO_2$ flux between anticyclonic and cyclonic eddies globally and regionally consistent with previous analyses. The driving mechanisms for these differences have been investigated in previous work. For example, Li et al. (2025) suggest that the competing changes in dissolved inorganic carbon and biological processes through eddy pumping contribute to the observed mesoscale eddy modification of the air-sea $CO_2$ flux. Additionally, Keppler et al. (2024) showed that the mesoscale modification of the air-sea $CO_2$ flux had significant seasonal variability in the Southern Ocean, indicating that underlying driving processes could vary throughout the individual eddies lifetime. Ford et al. (2023) showed that the changes in air-sea $CO_2$ flux in mesoscale eddies could be attributed to changes in the competing biological and physical processes. Although a comprehensive analysis of the driving mechanism is beyond the scope of this manuscript, the UEx-L-Eddies shows regional (Figure 8) and seasonal variability in the mesoscale eddy modification of the air-sea $CO_2$ flux (e.g. Figure S4 shows anticyclonic eddies have stronger uptake in winter). The underlying environmental parameters (e.g. SST, MLD) could therefore be used to investigate the driving mechanisms for these differences in the mesoscale modification.".*

Reviewer 2

This manuscript presents a global spatiotemporal database of ocean $CO_2$ covering the period 1993–2022, comprising 5,996 long-lived mesoscale eddies (3,244 anticyclonic and 2,752 cyclonic, each with a lifetime >1 year). Surface ocean $fCO_2$ (sw) was estimated using an improved neural-network approach (UExP-FNN-U), and air–sea $CO_2$ fluxes were calculated with FluxEngine, including a comprehensive uncertainty assessment. The study addresses a critical gap in air–sea carbon flux research by focusing on eddy-scale processes, combining satellite-derived eddy trajectories, reanalysis products, and machine-learning estimates. However, the current version of the manuscript spends too much effort interpreting the results and their implications, while paying insufficient attention to the methods and the demonstration of data reliability. Key details regarding data processing, quality control, and the neural-network training procedure are not clearly described. Moreover, the chlorophyll-based experiments and their comparison or validation are inadequate. I strongly recommend that the authors substantially strengthen the validation and methodological sections. Without a more rigorous demonstration of data and method reliability in capturing eddy features, the manuscript is not yet suitable for publication in Earth System Science Data.

*Response: We thank the reviewer for their appraisal of our manuscript. We have addressed all their comments below.*

**Major comments:**

1. The dataset is mainly based on an existing neural-network model, with no clear modification or adaptation for eddy-specific environments. The authors should clarify why this model is suitable for mesoscale eddy applications and provide targeted validation to demonstrate its reliability. How does this approach differ from other machine-learning models, such as Landschützer's framework? What specific features make it appropriate for eddy conditions? Without clear evidence that the model captures eddy-related processes or outperforms general models, its applicability to eddy environments remains unconvincing. Is there any better performance between this data product and other data products like Landschutzer, Gregor, Chau data product?

*Response: The neural network approach used in this study (UExP-FNN-U) to estimate the $fCO_{2\ (sw)}$ within the eddy (and surrounding environment) is a similar methodology but different architecture to the approaches referred to by the reviewer. The UExP-FNN-U is architecturally similar to the SOM-FNN approach (Landschützer et al., 2016), and the architecture is given in Ford et al. (2024). As highlighted by analyses within the Global Carbon Budget, the performance of all these approaches on the global scale is similar.*

*Although the UExP-FNN-U is not specifically modified for eddy conditions, the impact of the eddies on the physical and biological conditions will result in a $fCO_{2\ (sw)}$ representative of the eddy conditions. We present two lines of evidence to support this conclusion. Firstly, previous studies that employ similar methodologies, have used neural networks trained on all available in situ $fCO_{2\ (sw)}$ within their geographical regions of interest, and then applied these networks to study the eddy conditions (Ford et al., 2023; Li et al., 2025). This includes the study of Ford et al. (2023) (which is the basis for this new dataset) who performed an accuracy and precision analysis and showed that their neural network was able to capture the eddy $fCO_{2\ (sw)}$ conditions. Secondly, here, we have performed an evaluation within eddies to all available within-eddy in situ $fCO_{2\ (sw)}$ showing that the UExP-FNN-U has good accuracy (i.e bias close to 0) and a precision within the locations of eddies that is similar to the global accuracy and precision (as given in Ford et al., 2024). The within-eddy accuracy and precision estimates were discussed within the text at Lines 402-429 along with their comparison to the equivalent global assessment statistics of the UExP-FNN-U. In response to another of this reviewers' comments, we have now added the spatial residual maps and included seasonal splits for the within-eddy accuracy and precision estimates. These comparisons show the UExP-FNN-U is able to capture the eddy dynamics with little bias, and a precision consistent to the global training.*

*In hindsight we can see that some of our phrasing or naming in the text was a little unclear when we referred to accuracy and precision estimates, and so this may have confused the reader. Throughout the text we now clearly refer to all eddy specific performance as 'within the region of an eddy $fCO_{2\ (sw)}$ data performance' (or similar phrasing) so that it is clearly delineated from 'global $fCO_{2\ (sw)}$ data performance'.*

2. The comparison in Section 4.2 with previous studies (e.g., Guo & Timmermans 2024; Li et al. 2025; Keppler et al. 2024; Ford et al. 2023) is too general and does not sufficiently explain the sources of divergence in reported results. To strengthen the discussion, I recommend adding a comparative table or supplementary figure that systematically summarizes key methodological differences across studies—including eddy identification criteria, lifetime and radius thresholds, $fCO_2$ estimation methods, temporal coverage, spatial domain, and uncertainty treatment—and clarifying how these factors may drive discrepancies in both magnitude and interpretation. This would provide readers with a clearer comparative framework.

**Response:** *This is a good idea, thank you. We have now added a comparative table (Table 2) that summarises the differences between the studies compared to in Section 4.2. This new table is now referred to, and supports, the discussion in the main text.*

**Table 2: Summary of methodologies in previous studies used to estimate the eddy modification of the air-sea $CO_2$ flux. $pCO_{2\ (sw)}$ is the partial pressure of $CO_2$ in seawater.**

| | This study | Guo and Timmermans (2024) | Li et al. (2025) | Keppler et al. (2024) | Ford et al. (2023) |
|---|---|---|---|---|---|
| Eddy Dataset (or decomposition approach) | META 3.2 | Mesoscale signature decomposition | META 3.2 | META 3.2 | META 3.1exp |
| Lifetimes considered | > 1 year | N/A | >12 weeks | >=10 days | >1 year |
| Radius Threshold | No criteria | N/A | No criteria | >40km | No criteria |
| $fCO_{2\ (sw)}$ estimation method | Global $fCO_{2\ (sw)}$ neural network approach | Eddy resolving model | Regional $pCO_{2\ (sw)}$ neural network approach | In situ pH with neural network Total Alkalinity | Regional $pCO_{2\ (sw)}$ neural network-approach |
| Temporal Coverage | January 1993 to December 2022 | 1982 to 2000 | July 2002 to 1 January 2022 | April 2014 to February 2022 | July 2002 to December 2018 |
| Spatial Domain | Global | Global | Western Boundary Current (Kuroshio and Gulf Stream) | Southern Ocean | South Atlantic Ocean |
| Air sea $CO_2$ flux uncertainty treatment | Comprehensive uncertainty | N/A | $fCO_{2\ (sw)}$ and gas transfer considered | Standard error of observations | $fCO_{2\ (sw)}$ and gas transfer considered |

3. The authors should devote greater effort to demonstrating the reliability of the data rather than focusing excessively on result interpretation. The current evaluation of the $fCO_2$ neural network relies only on overall bias and RMSD, which may obscure regional or seasonal biases. A more thorough validation—such as stratified tests by region, season, eddy lifetime, eddy size, or chlorophyll level, and spatial maps of residuals—would better reveal systematic errors and the contribution of different eddy features. Such analyses are essential to improve uncertainty characterization and strengthen confidence in the derived air–sea $CO_2$ flux estimates.

**Response:** *We thank the reviewer for this suggestion. We have now included the requested spatial maps of residuals and the seasonal split of results. We have discussed above (in an earlier response) how the precision and accuracy assessments of the air-sea gas flux estimates are specific and valid for the region within and around each eddy, and this was supported by references and the eddy-local specific statistics we have provided. The chlorophyll-a data are climate data records resolved to the spatial resolution relevant to the mesoscale eddies and, as they are a climate data record, they have been extensively assessed and evaluated by the teams that produce them; our manuscript provides the references for these evaluations. Consequently, we feel that the evaluation is now very extensive and well supported by the literature. We thank the reviewer for the suggestions.*

*In producing the suggested spatial residual maps, we identified an error within our matching code for eddies close to the international dateline, and this has now been corrected. Fixing this, reduced the number of matches between the neural network estimated $fCO_{2\ (sw)}$ within eddies and the SOCAT in situ observations by ~700 for the anticyclonic eddies and ~400 for the cyclonic eddies. For anticyclonic eddies, the statistical comparisons between the neural network $fCO_{2\ (sw)}$ and in situ $fCO_{2\ (sw)}$ now includes 2082 matches, and for cyclonic the number of matches is 1376, therefore our statistics are still robust.*

*As requested, Figure 4 has been updated to now include residual maps, and the updated Figure 4 is displayed below in this document. The residual maps did not indicate any regional biases in the UExP-FNN-U $fCO_{2\ (sw)}$ within the eddies but did highlight that the majority of the in situ versus neural network estimated $fCO_{2\ (sw)}$ collocations occurred in the Northern Hemisphere where more in situ $fCO_{2\ (sw)}$ observations are made. This information has been added at Lines 263-272, which reads "The within eddy accuracy and precision estimates between the SOCAT in situ observations and the UExP-FNN-U $fCO_{2\ (sw)}$ showed good performance (Figure 4) similar to the results for the global scale in Ford et al. (2024) (weighted bias = -0.18 µatm, RMSD = 20.65, N = 18226 monthly 1 degree regions). For anticyclonic eddies, we observed a smaller weighted RMSD (precision) of 19.15 µatm (N=2082 monthly matches; Figure 4a). For cyclonic eddies we observed a lower RMSD of 16.49 µatm (N = 1376; Figure 4d). Both eddy types showed small weighted biases (accuracy) and therefore we consider the UExP-FNN-U generated $fCO_{2\ (sw)}$ within eddies to sufficiently represent the eddy $fCO_{2\ (sw)}$. The differences between the within-eddy UExP-FNN-U $fCO_{2\ (sw)}$ and in situ SOCAT observations did not indicate regional biases, but did show a spatial weighting to the Northern Hemisphere where more in situ $fCO_{2\ (sw)}$ are made (Bakker et al., 2016; Figure 4c, f).".*

*As requested, we have now included a new Figure (Figure 5) that splits the in situ versus UExP-FNN-U $fCO_{2\ (sw)}$ comparisons for the UExP-FNN-U into seasons. The new Figure 5 is displayed below in this document. Figure 5 shows that the UExP-FNN-U has good performance during winter and autumn but shows slightly larger weighted bias (i.e. slightly reduced accuracy) and RMSD (slightly reduced precision) during spring and autumn. Overall these comparisons showed no large seasonal biases in the UExP-FNN-U $fCO_{2\ (sw)}$ retrievals. This information has been added in Section 3.3, at Lines 273-277, which reads "Seasonally separating the collocated within eddy in situ observations shows that the UExP-FNN-U tended to show a small weighted bias (accuracy) and smaller RMSD (precision) during winter and autumn (Figure 5a,b,g,h) compared to spring and summer (Figure 5c,d,e,f). Although winter and autumn tended to have lower collocations between in situ SOCAT observations and the UExP-FNN-U $fCO_{2\ (sw)}$ (Figure 5). These seasonal comparisons further strengthen the accuracy and precision of the UExP-FNN-U $fCO_{2\ (sw)}$ and indicates no large seasonal biases.". Additionally within the discussion, we have also added text at Lines 409-411 which reads "Although, we did observe a slightly lower precision during the spring and summer, which could be due to the lack of a biological predictor (e.g chl-a) reducing the ability of the UExP-FNN-U to capture these dynamics (Ford et al., 2022) (Figure 5).".*

*These two Figures have also been replicated for the UExP-FNN-U with chl-a as a predictor and appear in the Supplementary as Figures S2 and S3. Figures S2 and S3 are also displayed below in this document. These comparisons highlighted that the secondary neural network with chl-a as a predictor showed improved accuracy and precision, and we noted that the spring and summer indicated greater performance improvements. We refer to these Figures within the discussion at Lines 422-427, where we have added new text which reads "This additional neural network showed similar but slightly improved precision (lower weighted RMSD) when compared to the in situ SOCAT observations, although to a lower number of data points (Figure S2; anticyclonic bias = -0.92 uatm, RMSD = 17.05 µatm, N = 1914; cyclonic bias = 0.05 µatm, RMSD = 14.31 µatm, N = 1272). In addition, the seasonal breakdown of the comparisons between the within eddy UExP-FNN-U with chl-a fCO₂ (sw) and the in situ fCO₂ (sw) showed an increase in the performance of this neural network during spring and summer, highlighting the improvements from chl-a being added as a predictor (Figure S3).".*

[Figure]

**Figure 4: (a) Comparison of the UExP-FNN-U fCO$_{2\ (sw)}$ to in situ SOCAT observations within anticyclonic eddies. Solid black line is the 1:1. Dashed line is the Type II linear regression. In text statistics are root mean square difference (RMSD), bias, slope and intercept of a Type II linear regression and number of matches (N). (b) same as (a) but showing the uncertainty on the fCO$_{2\ (sw)}$ (2 sigma; 95% confidence) as errorbars for anticyclonic eddies. (c) Difference between UExP-FNN-U fCO$_{2\ (sw)}$ to in situ SOCAT observations within anticyclonic eddies plotted as spatial residuals. (d, e and f) same as (a, b and c) for cyclonic eddies.**

[Figure]

**Figure 5:** (a) Comparison of the UExP-FNN-U fCO₂ (sw) to in situ SOCAT observations within anticyclonic eddies during winter. Solid black line is the 1:1. Dashed line is the Type II linear regression. In text statistics are root mean square difference (RMSD), bias, slope and intercept of a Type II linear regression and number of matches (N). (b) same as (a) but for cyclonic eddies in the winter. (c) and (d) same as (a) and (b) for spring. (e) and (f) same as (a) and (b) for summer. (g) and (h) same as (a) and (b) for autumn.

[Figure]

**Figure S2:** (a) Comparison of the neural network fCO₂ (sw) (with chl-a added as a predictor) to in situ SOCAT observations within anticyclonic eddies. Solid black line is the 1:1. Dashed line is the Type II linear regression. In text statistics are root mean square difference (RMSD), bias, slope and intercept of a Type II linear regression and number of matches (N). (b) same as (a) but showing the uncertainty on the neural network fCO₂ (sw) (2 sigma; 95% confidence) as errorbars for anticyclonic eddies. (c and d) same as (a and b) for cyclonic eddies.

[Figure]

**Figure S3: (a)** Comparison of the neural network fCO₂ (sw) (with chl-a added as a predictor) to in situ SOCAT observations within anticyclonic eddies during winter. Solid black line is the 1:1. Dashed line is the Type II linear regression. In text statistics are root mean square difference (RMSD), bias, slope and intercept of a Type II linear regression and number of matches (N). **(b)** same as (a) but for cyclonic eddies in the winter. **(c)** and **(d)** same as (a) and (b) for spring. **(e)** and **(f)** same as (a) and (b) for summer. **(g)** and **(h)** same as (a) and (b) for autumn.

4. Since ESSD focuses on data production and transparency, it would greatly help readers if the authors clearly present the framework of the neural-network architecture as well as the workflow of data processing, testing, and validation. A schematic of machine learning method and a flowchart of data production would substantially improve clarity and reproducibility.

*Response: As suggested, we have now added a new schematic that shows the data processing procedure applied within our study, which is now the new Figure 1. This new figure can be found below in this document, and it introduces the eddy tracking, the match up methodology and environmental datasets, and the generation of the air-sea gas flux data. The neural network approach (UExP-FNN-U) is explained in detail within a schematic in Ford et al. (2024), and the neural network training is consistent to this description. We have therefore not added this information to Figure 1, but we have included a summary of the neural network architecture, and the testing and validation approach used in our study. The updated text on the UExP-FNN-U can be found at Lines 124-135 which reads "The methods used are consistent with those in Ford et al (2024a), so only a summary of the method is provided here. The UExP-FNN-U is a two-step self-organising map (SOM) feed forward neural network (FNN) setup. The SOM splits the global ocean into 16 regions with a similar fCO₂ (sw), SST, SSS and MLD seasonal cycles. A FNN ensemble (10 FNNs for each region) was then trained with in situ monthly 1 degree fCO₂ (sw) observations from the Surface Ocean CO₂ Atlas (SOCAT; Bakker et al., 2016) that have been recalculated to a consistent temperature and depth dataset (Ford, Shutler, et al., 2024). The monthly 1 degree predictor variables of SST, SSS, MLD and the atmospheric dry mixing ratio of CO₂ (xCO₂ (atm)), and anomalies of each with respect to a long term monthly climatology were collocated to the in situ fCO₂ (sw). The FNNs consists of an input layer with nodes equal to the number of input predictors, a hidden layer with a varying number of nodes depending on a pretraining step and an output layer with a single node. The training data were split into a 95% training and validation dataset, and a 5% independent test randomly for each month ensuring the independent data were not clustered in one region. The UExP-FNN-U fCO₂ (sw) estimates are then typically used to estimate the global ocean CO₂ sink as described in Ford et al. (2024).".*

[Figure]

**Figure 1: Schematic showing the processing steps to estimate the air-sea CO₂ flux within long lived eddies (Blue box background). The pink background boxes indicate the analysis completed to evaluate the accuracy and precision of the dataset. In figure acronyms are: fugacity of CO₂ in seawater (fCO₂ (sw)), atmospheric dry mixing ratio of CO₂ (xCO₂ (atm)) and University of Exeter feed forward neural network with uncertainties (UExP-FNN-U).**

5. The analysis focuses only on long-lived eddies (lifetime > 1 year; radius > 30 km), which account for merely ~0.4% of all eddies in the AVISO data. The authors should justify this restrictive sampling choice. Because the vast majority of shorter-lived, smaller-scale eddies—particularly prevalent in western boundary currents and equatorial regions—are excluded, yet they may exert a substantial and possibly different cumulative influence on air–sea CO₂ fluxes. To avoid overgeneralization, I recommend that the authors (i) clearly state that their conclusions apply only to this subset of long-lived eddies, (ii) provide the eddy lifetime distribution and grouped statistics (e.g., sample size, mean radius, spatial coverage, flux contribution) across different lifetime classes, and (iii) if feasible, perform a simplified analysis on short-lived eddies, or otherwise explicitly discuss the likely direction and magnitude of biases introduced by their exclusion with reference to existing literature.

*Response: We have now added information on the reason for our focus of long-lived eddies, which occurs at Lines 101-105 that reads "The focus on these long-lived eddies was due to their presence likely exhibiting a larger influence on the air-sea CO₂ flux (e.g. Smith et al., 2023). Additionally, the selection was due to computational limitations in running the analysis for the extensive set of shorter lived eddies within the dataset. We are working to extend the analysis to shorter lived eddies but currently the focus remains on long lived eddies.".*

*We have not been able to run the analysis on shorter lived eddies due to the computational reasons highlighted above. However, we have now included a statement within the limitations that suggests the likely direction of the mesoscale eddy modification of the air-sea CO₂ flux. This text reads at Lines 518-524 as "The UEx-L-Eddies dataset focusses on larger, long-lived eddies (lifetimes greater than a year). This criteria will regionally exclude eddies within, for example, highly dynamic western boundary currents where shorter lived eddies often dominate (Figure 2c, d). Smith et al. (2023) however show that eddies with smaller radii generally have the same anomaly direction but with weaker magnitudes when compared to larger eddies. A previous study (Pegliasco et al., 2022) identified that the shorter lived eddies within the Mesoscale Eddy Product (the same product used within this study) generally have smaller radii then the*

*longer lived eddies. Therefore we would expect similar anomalies but of smaller magnitude when studying shorter lived eddies.".*

6. Some data-processing procedures are insufficiently described, which is not sufficient for a data-oriented paper. The authors note that on some days environmental data (e.g., CCI-SST) are missing due to the absence of defined eddy polygons, but the extent of this missing data has not been quantified, nor is it clear whether certain eddies were excluded or flagged. It is recommended to describe the treatment strategy for cases with substantial data loss (e.g., exclusion, interpolation, or flagging). If no exclusion was applied, the potential direction of bias should be discussed. This would ensure that database users can properly interpret and filter eddy time series that may be affected by data quality issues.

*Response: This is a good point. We have now added further details to data-processing procedures as highlighted by the reviewer. For the missing polygons, we have now quantified that the missing data affects a mean of 2% of an individual eddy daily time series, and these occur randomly throughout the timeseries. As the impact is randomly through the timeseries (and effects ~2%), the impact on the monthly median statistics is minimal. We have added this information at Lines 512-517 as "For some eddies the daily environmental data can have missing values even for complete coverage data (for example, the CCI-SST). These gaps stem from the META3.2 eddy trajectories dataset where the polygon to define the limits of the eddy does not form correctly, and therefore we were unable to extract values where the polygon was undefined. No exclusion or interpolation mechanism was implemented as these data gaps affect a mean of 2 % (maximum = 15 %) of an individual eddy daily timeseries, which occur randomly through the timeseries, and therefore the impact on the monthly median statistics are minimal.".*

7. The authors provide a second $fCO_2$ estimate that includes chl-a as an input, but this product is only available from 1997 onward and contains gaps in polar regions during winter. The manuscript currently mentions these limitations only briefly, without quantifying their impact. I recommend that the authors discuss how this affects the temporal and regional representativeness of the results; and (ii) present spatial difference maps or regional statistics comparing the estimates with and without chl-a, to allow readers to assess the role of biological factors in different oceanic regimes.

*Response: We thank the reviewer for their suggestion. We have now included an additional figure (Figure 9) into our discussion of the chl-a version of the UExP-FNN-U, which replicates Figure 8 for the chl-a version of the UExP-FNN-U but also shows the results of using the same eddy set with the UExP-FNN-U without the inclusion of chl-a in the neural network training. Figure 9 is shown below in this document and confirms that the air-sea $CO_2$ flux estimate is different between the two versions, but that all differences fall within the uncertainties. Notably the South Pacific and Southern Ocean show larger differences than the other regions. These details are now discussed in a new paragraph at Lines 430-437 which reads "The impact on the modification of the cumulative air-sea $CO_2$ flux by mesoscale eddies due to including chl-a within the UExP-FNN-U can be assessed by replicating Figure 8, but using the secondary $fCO_{2\ (sw)}$ and resulting air-sea $CO_2$ fluxes (Figure 9). Figure 9 shows the regional modification of the air-sea $CO_2$ fluxes by eddies where both neural network variants are able to estimate the $fCO_{2\ (sw)}$ (i.e we show a subset of the eddies in Figure 8). In all regions both neural networks retrieve a similar signature, but the chl-a version generally suggests a stronger enhancement (or weaker suppression) of the $CO_2$ sink compared to the UExP-FNN-U without chl-a. Notably the South Pacific Ocean and Southern Ocean show larger differences although in all cases these differences fall within the uncertainties. We therefore provide the secondary neural network to further aid in understanding the processes that are driving mesoscale eddy modification of the air-sea $CO_2$ fluxes.".*

[Figure]

**Figure 9:** (a) Ocean basins considered for further analysis, with a colour for each region. Regions follow the RECCAP2 ocean basin definition, but each basin was split at the Equator into North and South. North Indian Ocean was removed due to low number of eddies analysed. (b) Box plot showing the eddy modification of the air-sea $CO_2$ flux using the chl-a version of the UExP-FNN-U for the region shown with the arrow. Red line indicates the median, box indicates the 25th and 75th quartiles, whiskers extend from the 25th and 75th quartiles by 1.5 interquartile ranges. Circles indicate data considered outliers (greater than 1.5 interquartile ranges outside the 25th and 75th percentile). Dark red shading indicates the 1 sigma (~68% confidence) uncertainty on the median by propagating the air-sea $CO_2$ flux uncertainties using a Monte Carlo uncertainty propagation. Light red shading indicates the 2 sigma uncertainty on the median (~95% confidence). Blue line and shading indicates the same but for the UExP-FNN-U without chl-a. X-axis label shows number of eddy (N), the median modification with the 2 sigma uncertainty for the chl-a version of the UExP-FNN-U labelled with a C, and the UExP-FNN-U without chl-a labelled with a P. (c), (d), (e), (f), (g) same as (b) for their respective regions identified by the arrow. Basemap in (a) from Natural Earth v4.0.0 (https://www.naturalearthdata.com/).

8. The discussion on data reliability and methodological limitations is insufficient. The authors should explicitly identify the main sources of bias arising from both their approach and data, and propose strategies to improve robustness and reproducibility. For example, conducting additional machine-learning experiments to trace and quantify potential biases would help strengthen the credibility of the dataset.

***Response:*** *We have used an evidence-based approach to guide our discussion of the limitations. The uncertainty analysis presented within the manuscript shows that the largest contribution is from the $fCO_{2\ (sw)}$ component, and therefore the neural network estimates. We have discussed the potential limitations, and biases with the $fCO_{2\ (sw)}$ approach within the methodology and discussion. We highlight the exclusion of a representative biological parameter could affect the neural network, and therefore provide the second version of the neural network $fCO_{2\ (sw)}$ estimates that include chl-a. To assess biases in the neural network estimated $fCO_{2\ (sw)}$ we performed the statistical comparison to the in situ SOCAT observations within eddies, which has been expanded in response to an earlier comment by the reviewer. We have also now extended the evaluation of the chl-a neural network $fCO_{2\ (sw)}$ as also suggested by the reviewer.*

*In addressing the reviewers comment above, we produced a new figure which compared the change in the eddy flux compared to the surrounding water estimated from the two neural networks (Figure 9). These showed small differences in the regional modification by both anticyclonic and cyclonic, although on a smaller sample size due to limitations discussed with the chl-a neural network on Lines 155-159.*

*We therefore feel the data reliability and methodological limitations are now well discussed within the manuscript, and we thank the reviewer' for their earlier comments which have helped to strengthen this further.*

**Minor comments:**

1. I suggest adding a table summarizing the data sources and key characteristics of all variables used in this study (e.g., variable, units, period, resolution, product name, and references). Such a table would improve clarity, allow readers to quickly assess the datasets employed.

*Response: We have now added a new table that summarises the environmental input parameters that are considered within the analysis. Table 1 is on Line 95 of the manuscript and shown below this response.*

**Table 1: Summary of the environmental datasets and in situ observations collocated with the long lived mesoscale eddies.**

| Parameter | Units | Dataset | Temporal Resolution | Spatial Resolution | Reference |
|---|---|---|---|---|---|
| Sea surface temperature | Kelvin | ESA CCI-SST v3.0 | Daily | ~5km (0.05 degree) | (Embury et al., 2024; Good & Embury, 2024) |
| Sea surface salinity | Psu | CMEMS GLORYS12V1 | Daily | ~9km (0.08 degree) | (CMEMS, 2021; Jean-Michel et al., 2021) |
| Mixed layer depth | m | CMEMS GLORYS12V1 | Daily | ~9km (0.08 degree) | (CMEMS, 2021; Jean-Michel et al., 2021) |
| Chlorophyll-a | mg m$^{-3}$ | OC-CCI v6 | Daily | 4km | (Sathyendranath et al., 2019, 2023) |
| Wind speed | m s$^{-1}$ | CCMP v3.1 | 6 hourly | ~25km (0.25 degree) | (Mears et al., 2022; Remote Sensing Systems et al., 2022) |
| Sea level pressure | hPa | ERA5 | Monthly | ~25km (0.25 degree) | (Hersbach et al., 2019, 2020) |
| $xCO_{2 (atm)}$ | ppm | NOAA-GML | Monthly | ~100km (1 degree) | (Lan et al., 2023) |
| $fCO_{2 (sw)}$ | µatm | Recalculated SOCAT | Individual cruise observations | N/A | (Bakker et al., 2016; Ford, Shutler, et al., 2024) |

2. Regarding $xCO_2$ (MBL), please clarify how the meridional band product was mapped onto the 1° field (e.g., through band replication, interpolation, or another approach). Providing this detail would improve the transparency of the data processing procedure.

*Response: We have now added the information on the $xCO_{2 (atm)}$, whereby the zonal marine boundary layer $xCO_{2 (atm)}$ was replicated for each longitude. This information has been added at Lines 140-143 as "These $xCO_{2 (atm)}$ fields were produced by calculating the monthly average of the $xCO_{2 (atm)}$ for each latitude (~2.5 degree spacing), which were then interpolated to 1 degree and replicated for each 1 degree*

*longitude. A distance weighted mean of the nearest four pixels taken at the mean (centre) position of each eddy was used to estimate the monthly $xCO_{2\,(atm)}$.".*

3. The dataset spans 1993–2022. It is recommended that the authors at least comment on decadal-scale variations. For example, they could assess whether the impact of eddies on air–sea fluxes remained stable during the 1990s, 2000s, and 2010s, or if notable changes occurred. Even if a full trend analysis cannot be conducted in the main text, it would be helpful to provide simple decade-wise statistics (e.g., median changes per decade) in the supplementary materials to give readers an initial view of long-term evolution.

*Response: We have now added a new supplementary table (Table S1) that provides the median eddy modification for the periods 1993-2000, 2000-2010 and 2010-2020. Table S1 is displayed below this response. This analysis suggest that the anticyclonic eddies are becoming an increasing enhancer of the $CO_2$ sink, and the cyclonic eddy suppression is reducing, but the differences are within the reported uncertainties. We have now mentioned the analysis within the main text at Lines 384-387 which reads "These data allow a range of analyses to be conducted, for example, in this study, we show how the mesoscale modification of the air-sea $CO_2$ flux can be determined from these data regionally (Figure 7; Figure 8) or could be evaluated through time (e.g. Table S1 provides global decadal median mesoscale modifications suggesting an increasing enhancement of the $CO_2$ sink).".*

**Table S1: Median decadal eddy modification of the air-sea $CO_2$ flux for both cyclonic and anticyclonic eddies. Uncertainties are the 95 % confidence interval of the propagated uncertainties. Number in square brackets indicates the number of eddies considered. Note eddies that form in one period and dissipate in another are considered in both periods.**

| Time period | Anticyclonic eddy modification of air-sea $CO_2$ flux | Cyclonic eddy modification of air-sea $CO_2$ flux |
|---|---|---|
| 1993 to 2000 | -2.57 ± 6.23 % [753] | 1.66 ± 5.95 % [617] |
| 2000 to 2010 | -4.28 ± 4.53 % [1321] | 0.74 ± 4.32 % [1119] |
| 2010 to 2020 | -5.48 ± 3.62 % [1482] | 0.30 ± 3.60 % [1283] |

4. The manuscript states that SOCAT data have been gridded (monthly 1°) and used for model training and testing, but the strategy for splitting the training and test sets is not clearly described. It is recommended that the authors provide details on the partitioning method employed, such as leave-one-time-out or leave-one-location-out cross-validation, or any other approach used, to clarify the reliability and independence of the model evaluation.

*Response: We have now added the information on the independent test data withheld from the UExP-FNN-U which is described in Ford et al. (2024). A 5% independent test dataset is withheld from each month of the full UExP-FNN-U training dataset and ensuring that the data are not clustered in one region. The remaining data (95%) are randomly split between training and validation for each of the 10 FNN ensemble members. The independent test data remains completely independent until the accuracy and precision are assessed. This information has been added at Lines 130-135 which reads "The FNNs consists of an input layer with nodes equal to the number of input predictors, a hidden layer with a varying number of nodes depending on a pretraining step and an output layer with a single node. The training data were split into a 95% training and validation dataset, and a 5% independent test randomly for each month ensuring the*

*independent data were not clustered in one region. The UExP-FNN-U fCO$_{2\ (sw)}$ estimates are then typically used to estimate the global ocean CO$_2$ sink as described in Ford et al. (2024).".*

5. Figure 1 currently only shows the eddy formation points and cumulative flux scatter. It is recommended to also overlay the eddy occurrence frequency or sample density at each grid point (or provide this in a supplementary figure). This would help assess whether certain large flux values are driven by a few exceptionally large or long-lived eddies.

***Response:*** *We have updated Figure 2 based on the recommendations of both reviewers, where we have added two new subplots that show the frequency of long-lived eddies formation compared to all eddy formations within 1 degree regions. The updated Figure 2 can be found on Page 3 of this document. We have also included the new supplementary Figure S1 that replicates Figure 2 but normalises the cumulative air-sea CO$_2$ flux to the eddy lifetime in days (giving a CO$_2$ flux in Tg C d$^{-1}$). This new supplementary Figure S1 is shown below in this document.*

[Figure]

**Figure S1: (a) The cumulative air-sea CO₂ flux into the anticyclonic eddies normalised by eddy lifetime in days where the scatter points are plotted at the formation location of each eddy. (b) same as (a) but for cyclonic eddies.**

6. Although the manuscript notes that anticyclonic eddies enhance $CO_2$ uptake while cyclonic eddies reduce it, it is recommended that the authors further quantify the asymmetry between the two eddy types. For instance, seasonal or regional statistics of flux differences could be provided, along with a discussion of potential physical drivers (e.g., temperature, stratification) and biological drivers (e.g., productivity).

*Response: We thank the reviewer for the suggestion, and we have now provided a synopsis of the seasonal variability in the mesoscale modification of the air-sea $CO_2$ flux in the supporting information (Figure S4). Figure S4 is displayed below this response in this document. The regional synopsis of the mesoscale eddy modification of the air-sea $CO_2$ fluxes was presented in the original manuscript (Figure 7, Figure 8 and discussed in Section 4.2), which has been extended following comments by the reviewer. We have now added a paragraph that explores the potential mechanisms driving these differences which are supported by the previous research. A comprehensive analysis of the driving mechanisms is beyond the*

*scope of this manuscript. The new paragraph at Lines 494-505 reads "The UEx-L-Eddies identifies differences in the mesoscale eddy modification of the cumulative air-sea $CO_2$ flux between anticyclonic and cyclonic eddies globally and regionally consistent with previous analyses. The driving mechanisms for these differences have been investigated in previous work. For example, Li et al. (2025) suggest that the competing changes in dissolved inorganic carbon and biological processes through eddy pumping contribute to the observed mesoscale eddy modification of the air-sea $CO_2$ flux. Additionally, Keppler et al. (2024) showed that the mesoscale modification of the air-sea $CO_2$ flux had significant seasonal variability in the Southern Ocean, indicating that underlying driving processes could vary throughout the individual eddies lifetime. Ford et al. (2023) showed that the changes in air-sea $CO_2$ flux in mesoscale eddies could be attributed to changes in the competing biological and physical processes. Although a comprehensive analysis of the driving mechanism is beyond the scope of this manuscript, the UEx-L-Eddies shows regional (Figure 8) and seasonal variability in the mesoscale eddy modification of the air-sea $CO_2$ flux (e.g. Figure S4 shows anticyclonic eddies have stronger uptake in winter). The underlying environmental parameters (e.g. SST, MLD) could therefore be used to investigate the driving mechanisms for these differences in the mesoscale modification.".*

[Figure]

**Figure S4: (a) Box plot showing the eddy modification of the air-sea $CO_2$ flux during winter using the UExP-FNN-U. Red line indicates the median, box indicates the 25th and 75th quartiles, whiskers extend from the 25[th] and 75[th] quartiles by 1.5 interquartile ranges. Circles indicate data considered outliers. Dark red shading indicates the 1 sigma (~68% confidence) uncertainty on the median by propagating the air-sea $CO_2$ flux uncertainties using a Monte Carlo uncertainty propagation. Light red shading indicates the 2 sigma uncertainty on the median (~95% confidence). X-axis label shows number of eddies (N), the median modification with the 2 sigma uncertainty. (b) same as (a) but for spring. (c) same as (a) but for summer. (d) same as (a) for autumn.**

7. This study frequently refers to the "weighted mean," but the weighting scheme is not clearly specified (e.g., latitude, area, or distance weighting). It is recommended that the authors add a statement clarifying how the weights are calculated.

***Response:*** *We have now clarified that the weighted means are calculated based on distance to mean location of the eddy and this is now stated at Lines 141-144 and 193-195.*

8. Figure 1 shows that many eddy formation points are located near the coast or continental shelf, but the manuscript does not specify how eddy polygons overlapping with land on a given day are handled. This could affect the extraction of sea surface variables and flux estimates. It is recommended that the authors clarify the treatment in the Methods section (e.g., applying a land mask to retain only the ocean portion, or excluding eddy days where the overlap with land exceeds a certain threshold). If no such treatment was applied, it would be helpful to provide in the supplementary materials statistics on the number or fraction of eddies overlapping with land.

*Response: Within Figure 2 some of the eddies do form close to the coastal region, however none of these overlap with the land. The eddy trajectories and the polygons for the eddy location are estimated from satellite altimetry sea level heights which do not have land values. Therefore these features cannot overlap land. We have now added this information at Lines 106-107 which reads "For each eddy trajectory, a daily position was provided along with a polygon shape that estimates the eddy shape and size from the altimetry-based data which can not overlap with land.".*

9. This study employs Type II regression to compare the NN and SOCAT data, but it does not briefly explain why Type II rather than ordinary least squares (OLS) regression was chosen. It is recommended to add 1–2 sentences in the statistical comparison section to justify this choice (e.g., Type II regression is more appropriate when both the independent and dependent variables contain measurement errors) and to cite relevant references.

*Response: We have now added a sentence to describe why a Type II regression was used. This sentence reads at Lines 167-169 as: "A Type II linear regression was used as uncertainties are presented within both the in situ and neural network $fCO_{2\ (sw)}$ (Laws, 1997; York et al., 2004).".*

**References**

Bakker, D. C. E., Pfeil, B., Landa, C. S., Metzl, N., O'Brien, K. M., Olsen, A., et al. (2016). A multi-decade record of high-quality $fCO_2$ data in version 3 of the Surface Ocean $CO_2$ Atlas (SOCAT). *Earth System Science Data*, *8*(2), 383–413. https://doi.org/10.5194/essd-8-383-2016

CMEMS. (2021). Copernicus Marine Modelling Service global ocean physics reanalysis product (GLORYS12V1). *Copernicus Marine Modelling Service [Data Set]*. https://doi.org/10.48670/moi-00021

Embury, O., Merchant, C. J., Good, S. A., Rayner, N. A., Høyer, J. L., Atkinson, C., et al. (2024). Satellite-based time-series of sea-surface temperature since 1980 for climate applications. *Scientific Data*, *11*(1), 326. https://doi.org/10.1038/s41597-024-03147-w

Ford, D. J., Tilstone, G. H., Shutler, J. D., & Kitidis, V. (2022). Derivation of seawater $pCO_2$ from net community production identifies the South Atlantic Ocean as a $CO_2$ source. *Biogeosciences*, *19*(1), 93–115. https://doi.org/10.5194/bg-19-93-2022

Ford, D. J., Tilstone, G. H., Shutler, J. D., Kitidis, V., Sheen, K. L., Dall'Olmo, G., & Orselli, I. B. M. (2023). Mesoscale Eddies Enhance the Air-Sea $CO_2$ Sink in the South Atlantic Ocean. *Geophysical Research Letters*, *50*(9), e2022GL102137. https://doi.org/10.1029/2022GL102137

Ford, D. J., Blannin, J., Watts, J., Watson, A. J., Landschützer, P., Jersild, A., & Shutler, J. D. (2024). A Comprehensive Analysis of Air-Sea $CO_2$ Flux Uncertainties Constructed From Surface Ocean Data Products. *Global Biogeochemical Cycles*, *38*(11), e2024GB008188. https://doi.org/10.1029/2024GB008188

Ford, D. J., Shutler, J. D., Ashton, I., Sims, R. P., & Holding, T. (2024). Reanalysed (depth and temperature consistent) Surface Ocean $CO_2$ Atlas (SOCAT) version 2024 (v1.1) (Version v1.1) [Data set]. Zenodo. https://doi.org/10.5281/ZENODO.13284017

Good, S. A., & Embury, O. (2024). ESA Sea Surface Temperature Climate Change Initiative (SST_cci): Level 4 Analysis product, version 3.0 [Application/xml]. NERC EDS Centre for Environmental Data Analysis [dataset]. https://doi.org/10.5285/4A9654136A7148E39B7FEB56F8BB02D2

Guo, Y., & Timmermans, M. (2024). The Role of Ocean Mesoscale Variability in Air-Sea $CO_2$ Exchange: A Global Perspective. *Geophysical Research Letters*, *51*(10), e2024GL108373. https://doi.org/10.1029/2024GL108373

Hersbach, H., Bell, B., Berrisford, P., Biavati, G., Horányi, A., Muñoz Sabater, J., Nicolas, J., et al. (2019). ERA5 monthly averaged data on single levels from 1979 to present. *Copernicus Climate Change Service (C3S) Climate Data Store (CDS) [Dataset]*. https://doi.org/10.24381/cds.f17050d7

Hersbach, H., Bell, B., Berrisford, P., Hirahara, S., Horányi, A., Muñoz-Sabater, J., et al. (2020). The ERA5 global reanalysis. *Quarterly Journal of the Royal Meteorological Society*, *146*(730), 1999–2049. https://doi.org/10.1002/qj.3803

Jean-Michel, L., Eric, G., Romain, B.-B., Gilles, G., Angélique, M., Marie, D., et al. (2021). The Copernicus Global 1/12° Oceanic and Sea Ice GLORYS12 Reanalysis. *Frontiers in Earth Science*, *9*(July), 1–27. https://doi.org/10.3389/feart.2021.698876

Keppler, L., Eddebbar, Y. A., Gille, S. T., Guisewhite, N., Mazloff, M. R., Tamsitt, V., et al. (2024). Effects of Mesoscale Eddies on Southern Ocean Biogeochemistry. *AGU Advances*, *5*(6), e2024AV001355. https://doi.org/10.1029/2024AV001355

Lan, X., Tans, P., Thoning, K., & NOAA Global Monitoring Laboratory. (2023). NOAA Greenhouse Gas Marine Boundary Layer Reference - CO2. [Data set]. NOAA GML. https://doi.org/10.15138/DVNP-F961

Landschützer, P., Gruber, N., & Bakker, D. C. E. (2016). Decadal variations and trends of the global ocean carbon sink. *Global Biogeochemical Cycles*, *30*(10), 1396–1417. https://doi.org/10.1002/2015GB005359

Laws, E. A. (1997). *Mathematical methods for oceanographers: an introduction*. New York: Wiley.

Li, X., Gan, B., Zhang, Z., Cao, Z., Qiu, B., Chen, Z., & Wu, L. (2025). Oceanic uptake of $CO_2$ enhanced by mesoscale eddies. *Science Advances*, *11*(24), eadt4195. https://doi.org/10.1126/sciadv.adt4195

Mears, C., Lee, T., Ricciardulli, L., Wang, X., & Wentz, F. (2022). Improving the Accuracy of the Cross-Calibrated Multi-Platform (CCMP) Ocean Vector Winds. *Remote Sensing*, *14*(17), 4230. https://doi.org/10.3390/rs14174230

Pegliasco, C., Delepoulle, A., Mason, E., Morrow, R., Faugère, Y., & Dibarboure, G. (2022). META3.1exp: a new global mesoscale eddy trajectory atlas derived from altimetry. *Earth System Science Data*, *14*(3), 1087–1107. https://doi.org/10.5194/essd-14-1087-2022

Remote Sensing Systems, Mears, C., Lee, T., Ricciardulli, L., Wang, X., & Wentz, F. (2022). RSS Cross-Calibrated Multi-Platform (CCMP) 6-hourly ocean vector wind analysis on 0.25 deg grid, Version 3.0 [Data set]. Santa Rosa, CA, USA: Remote Sensing Systems [dataset]. https://doi.org/10.56236/rss-uv6h30

Sathyendranath, S., Brewin, R. J. W., Brockmann, C., Brotas, V., Calton, B., Chuprin, A., et al. (2019). An ocean-colour time series for use in climate studies: The experience of the ocean-colour climate change initiative (OC-CCI). *Sensors*, *19*(19). https://doi.org/10.3390/s19194285

Sathyendranath, S., Jackson, T., Brockmann, C., Brotas, V., Calton, B., Chuprin, A., et al. (2023). ESA Ocean Colour Climate Change Initiative (Ocean_Colour_cci): Version 6.0, 4km resolution data [Data set]. NERC EDS Centre for Environmental Data Analysis [dataset]. https://doi.org/10.5285/5011D22AAE5A4671B0CBC7D05C56C4F0

Smith, T. G., Nicholson, S. -A., Engelbrecht, F. A., Chang, N., Mongwe, N. P., & Monteiro, P. M. S. (2023). The Heat and Carbon Characteristics of Modeled Mesoscale Eddies in the South−East Atlantic Ocean. *Journal of Geophysical Research: Oceans*, *128*(12), e2023JC020337. https://doi.org/10.1029/2023JC020337

York, D., Evensen, N. M., Martínez, M. L., & De Basabe Delgado, J. (2004). Unified equations for the slope, intercept, and standard errors of the best straight line. *American Journal of Physics*, *72*(3), 367–375. https://doi.org/10.1119/1.1632486